# SegFormer: Simple and Efficient Design for Semantic Segmentation with Transformers

**Enze Xie**[1]  **Wenhai Wang**[2]  **Zhiding Yu**[3*] **Anima Anandkumar**[3,4]  **Jose M. Alvarez**[3]  **Ping Luo**[1]

[1]The University of Hong Kong  [2]Nanjing University  [3]NVIDIA  [4]Caltech

xieenze@hku.hk, wangwenhai362@163.com,
{zhidingy,josea,aanandkumar}@nvidia.com, pluo@cs.hku.hk

## Abstract

We present SegFormer, a simple, efficient yet powerful semantic segmentation framework which unifies Transformers with lightweight multilayer perceptron (MLP) decoders. SegFormer has two appealing features: 1) SegFormer comprises a novel hierarchically structured Transformer encoder which outputs multiscale features. It does not need positional encoding, thereby avoiding the interpolation of positional codes which leads to decreased performance when the testing resolution differs from training. 2) SegFormer avoids complex decoders. The proposed MLP decoder aggregates information from different layers, and thus combining both local attention and global attention to render powerful representations. We show that this simple and lightweight design is the key to efficient segmentation on Transformers. We scale our approach up to obtain a series of models from SegFormer-B0 to SegFormer-B5, reaching significantly better performance and efficiency than previous counterparts. For example, SegFormer-B4 achieves 50.3% mIoU on ADE20K with 64M parameters, being $5\times$ smaller and 2.2% better than the previous best method. Our best model, SegFormer-B5, achieves 84.0% mIoU on Cityscapes validation set and shows excellent zero-shot robustness on Cityscapes-C. Code is available at: github.com/NVlabs/SegFormer.

## 1  Introduction

Semantic segmentation is a fundamental task in computer vision and enables many downstream applications. It is related to image classification since it produces per-pixel category prediction instead of image-level prediction. This relationship is pointed out and systematically studied in a seminal work [1], where the authors used fully convolutional networks (FCNs) for semantic segmentation tasks. Since then, FCN has inspired many follow-up works and has become a predominant design choice for dense prediction.

Since there is a strong relation between classification and semantic segmentation, many state-of-the-art semantic segmentation frameworks are variants of popular architectures for image classification on ImageNet. Therefore, designing backbone architectures has remained an active area

Figure 1: **Performance *vs.* model efficiency on ADE20K.** SegFormer achieves a new state-of-the-art 51.0% mIoU while being significantly more efficient than previous methods.

*Corresponding authors: Zhiding Yu and Ping Luo

35th Conference on Neural Information Processing Systems (NeurIPS 2021).

in semantic segmentation. Indeed, starting from
early methods using VGGs [1, 2], to the latest methods with significantly deeper and more powerful backbones [3], the evolution of backbones has dramatically pushed the performance boundary of semantic segmentation. Besides backbone architectures, another line of work formulates semantic segmentation as a structured prediction problem, and focuses on designing modules and operators, which can effectively capture contextual information. A representative example in this area is dilated convolution [4, 5], which increases the receptive field by "inflating" the kernel with holes.

Witnessing the great success in natural language processing (NLP), there has been a recent surge of interest to introduce Transformers to vision tasks. Dosovitskiy et al. [6] proposed vision Transformer (ViT) for image classification. Following the Transformer design in NLP, the authors split an image into multiple linearly embedded patches and feed them into a standard Transformer with positional embeddings (PE), leading to an impressive performance on ImageNet. In semantic segmentation, Zheng et al. [7] proposed SETR to demonstrate the feasibility of using Transformers in this task.

SETR adopts ViT as a backbone and incorporates several CNN decoders to enlarge feature resolution. Despite the good performance, ViT has two important limitations: 1) ViT outputs single-scale low-resolution features instead of multi-scale ones, and 2) it has very high computational cost on large images. To address these limitations, Wang et al. [8] proposed a pyramid vision Transformer (PVT), a natural extension of ViT with pyramid structures for dense prediction. PVT shows considerable improvements over the ResNet counterpart on object detection and semantic segmentation. However, together with other emerging methods such as Swin Transformer [9] and Twins [10], these methods mainly consider the design of the Transformer encoder, neglecting the contribution of the decoder for further improvements.

This paper introduces SegFormer, a cutting-edge Transformer framework for semantic segmentation that jointly considers efficiency, accuracy, and robustness. In contrast to previous methods, our framework redesigns both the encoder and the decoder. The key novelties of our approach are:

- A novel positional-encoding-free and hierarchical Transformer encoder.

- A lightweight All-MLP decoder design that yields a powerful representation without complex and computationally demanding modules.

- As shown in Figure 1, SegFormer sets new a state-of-the-art in terms of efficiency, accuracy and robustness in three publicly available semantic segmentation datasets.

First, the proposed encoder avoids interpolating positional codes when performing inference on images with resolutions different from the training one. As a result, our encoder can easily adapt to arbitrary test resolutions without impacting the performance. In addition, the hierarchical part enables the encoder to generate both high-resolution fine features and low-resolution coarse features, this is in contrast to ViT that can only produce single low-resolution feature maps with fixed resolutions. Second, we propose a lightweight MLP decoder where the key idea is to take advantage of the Transformer-induced features where the attentions of lower layers tend to stay local, whereas the ones of the highest layers are highly non-local. By aggregating the information from different layers, the MLP decoder combines both local and global attention. As a result, we obtain a simple and straightforward decoder that renders powerful representations.

We demonstrate the advantages of SegFormer in terms of model size, run-time, and accuracy on three publicly available datasets: ADE20K, Cityscapes, and COCO-Stuff. On Citysapces, our lightweight model, SegFormer-B0, without accelerated implementations such as TensorRT, yields 71.9% mIoU at 48 FPS, which, compared to ICNet [11], represents a relative improvement of 60% and 4.2% in latency and performance, respectively. Our largest model, SegFormer-B5, yields 84.0% mIoU, which represents a relative 1.8% mIoU improvement while being $5 \times$ faster than SETR [7]. On ADE20K, this model sets a new state-of-the-art of 51.8% mIoU while being $4 \times$ smaller than SETR. Moreover, our approach is significantly more robust to common corruptions and perturbations than existing methods, therefore being suitable for safety-critical applications. Code will be publicly available.

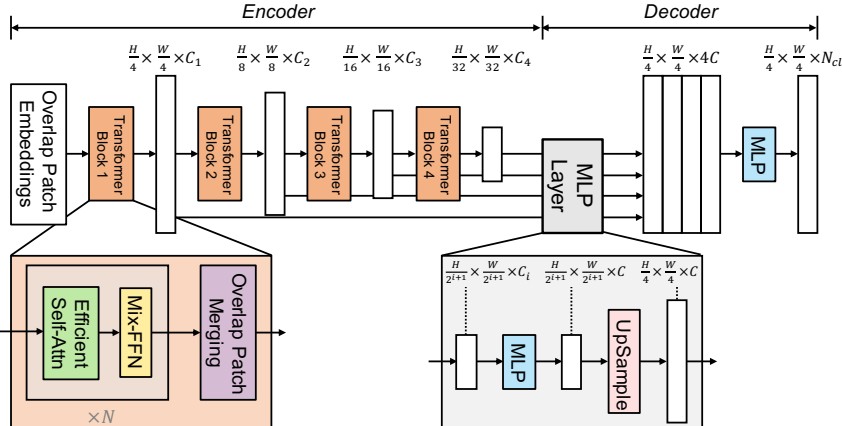

Figure 2: **The proposed SegFormer framework** consists of two main modules: A hierarchical Transformer encoder to extract coarse and fine features; and a lightweight All-MLP decoder to directly fuse these multi-level features and predict the semantic segmentation mask. "FFN" indicates feed-forward network.

## 2 Related Work

**Semantic Segmentation.** Semantic segmentation can be seen as an extension of image classification from image level to pixel level. In the deep learning era [12–14], FCN [1] is the fundamental work of semantic segmentation, which is a fully convolution network that performs pixel-to-pixel classification in an end-to-end manner. After that, researchers focused on improving FCN from different aspects such as: enlarging the receptive field [15–17, 5, 2, 4, 18]; refining the contextual information [19–27]; introducing boundary information [28–35]; designing various attention modules [36–44]; or using AutoML technologies [45–49]. These methods significantly improve semantic segmentation performance at the expense of introducing many empirical modules, making the resulting framework computationally demanding and complicated. More recent methods have proved the effectiveness of Transformer-based architectures for semantic segmentation [7, 44]. However, these methods are still computationally demanding.

**Transformer backbones**. ViT [6] is the first work to prove that a pure Transformer can achieve state-of-the-art performance in image classification. ViT treats each image as a sequence of tokens and then feeds them to multiple Transformer layers to make the classification. Subsequently, DeiT [50] further explores a data-efficient training strategy and a distillation approach for ViT. More recent methods such as T2T ViT [51], CPVT [52], TNT [53], CrossViT [54] and LocalViT [55] introduce tailored changes to ViT to further improve image classification performance.

Beyond classification, PVT [8] is the first work to introduce a pyramid structure in Transformer, demonstrating the potential of a pure Transformer backbone compared to CNN counterparts in dense prediction tasks. After that, methods such as Swin [9], CvT [56], CoaT [57], LeViT [58] and Twins [10] enhance the local continuity of features and remove fixed size position embedding to improve the performance of Transformers in dense prediction tasks.

**Transformers for specific tasks**. DETR [50] is the first to use Transformers for end-to-end object detection framework without non-maximum suppression (NMS). Other works have also used Transformers in tasks such as tracking [59, 60], super-resolution [61], re-id [62], colorization [63], retrieval [64] and multi-modal learning [65, 66]. For semantic segmentation, SETR [7] adopts ViT [6] as a backbone to extract features, achieving impressive performance. However, these Transformer-based methods have very low efficiency and, thus, difficult to deploy in real-time applications.

## 3 Method

As depicted in Figure 2, SegFormer consists of two main modules: (1) a hierarchical Transformer encoder; and (2) a lightweight All-MLP decoder to predict the final mask. Given an image with size $H \times W \times 3$, we first divide it into patches of size $4 \times 4$. Unlike ViT which uses $16 \times 16$, using fine-grained patches favors semantic segmentation. Second, we use these patches as input to the hierarchical Transformer encoder to get multi-level features with resolution {1/4, 1/8, 1/16, 1/32}

of the original image. We then pass these multi-level features to the All-MLP decoder to predict the segmentation mask with a $\frac{H}{4} \times \frac{W}{4} \times N_{cls}$ resolution, where $N_{cls}$ is the number of categories. In the remainder of this section, we first detail the proposed encoder and decoder designs and then summarize the main differences of our approach compared to SETR.

## 3.1 Hierarchical Transformer Encoder

We design a series of Mix Transformer encoders (MiT), MiT-B0 to MiT-B5, with the same architecture but different sizes. On top of the hierarchical architecture and efficient self-attention module in PVT [8], we further propose several novel features including **overlapped patch merging** and **positional-encoding-free design** which will be shown to greatly benefit the segmentation tasks.

**Hierarchical Feature Representation.** Unlike ViT [6], our encoder generates multi-level multi-scale features given an input image. These features provide both high-resolution coarse features and low-resolution fine-grained features that boost the performance of semantic segmentation. Specifically, given an input image with size $H \times W \times 3$, we perform patch merging to obtain a hierarchical feature map $F_i$ with a resolution of $\frac{H}{2^{i+1}} \times \frac{W}{2^{i+1}} \times C_i$, where $i \in \{1, 2, 3, 4\}$, and $C_{i+1}$ is larger than $C_i$.

**Efficient Self-Attention.** A major bottleneck of the above hierarchical feature representation is the quadratic self-attention complexity with long sequence inputs from higher resolution features. Recall that in the original multi-head self-attention, each of the heads $Q, K, V$ have the same dimensions $N \times C$, where $N = H \times W$ is the length of the sequence, the self-attention is estimated as:

$$\text{Attention}(Q, K, V) = \text{Softmax}(\frac{QK^{\top}}{\sqrt{d_{head}}})V. \tag{1}$$

We instead adopt the sequence reduction process introduced in [8]. This process uses a reduction ratio $R$ to reduce the length of the sequence of as follows:

$$\begin{aligned} \hat{K} &= \text{Reshape}(\frac{N}{R}, C \cdot R)(K) \\ K &= \text{Linear}(C \cdot R, C)(\hat{K}), \end{aligned} \tag{2}$$

where $K$ is the sequence to be reduced, $\text{Reshape}(\frac{N}{R}, C \cdot R)(K)$ refers to reshape $K$ to the one with shape of $\frac{N}{R} \times (C \cdot R)$, and $\text{Linear}(C_{in}, C_{out})(\cdot)$ refers to a linear layer taking a $C_{in}$-dimensional tensor as input and generating a $C_{out}$-dimensional tensor as output. Therefore, the new $K$ has dimensions $\frac{N}{R} \times C$. As a result, the complexity of the self-attention mechanism is reduced from $O(N^2)$ to $O(\frac{N^2}{R})$. In our experiments, we set $R$ to [64, 16, 4, 1] from stage-1 to stage-4.

**Overlapped Patch Merging.** Given an image patch, the patch merging process used in ViT, unifies a $N \times N \times 3$ patch into a $1 \times 1 \times C$ vector. This can easily be extended to unify a $2 \times 2 \times C_i$ feature path into a $1 \times 1 \times C_{i+1}$ vector to obtain hierarchical feature maps. Using this, we can shrink our hierarchical features from $F_1$ ($\frac{H}{4} \times \frac{W}{4} \times C_1$) to $F_2$ ($\frac{H}{8} \times \frac{W}{8} \times C_2$), and then iterate for any other feature map in the hierarchy. This process was initially designed to combine non-overlapping image or feature patches. Therefore, it fails to preserve the local continuity around those patches. Instead, we use an overlapping patch merging process. To this end, we define $K$, $S$, and $P$, where $K$ is the patch size, $S$ is the stride between two adjacent patches, and $P$ is the padding size. In our experiments, we set $K = 7$, $S = 4$, $P = 3$ ,and $K = 3$, $S = 2$, $P = 1$ to perform overlapping patch merging to produces features with the same size as the non-overlapping process. Similar to the original patch embedding in ViT [6], this operation can be implemented by "nn.Conv2D" in PyTorch.

**Positional-Encoding-Free Design.** The resolution of the PE in ViT is fixed. One thus needs to interpolate the PE when the test resolution differs from training. This leads to the drop of accuracy, which is undesirable since the resolution mismatch is common in semantic segmentation. We instead introduce Mix-FFN where we consider the effect of zero padding to the leak location information [67] by directly using a $3 \times 3$ Conv in the feed-forward network (FFN). Mix-FFN is formulated as:

$$\mathbf{x}_{out} = \text{MLP}(\text{GELU}(\text{Conv}_{3 \times 3}(\text{MLP}(\mathbf{x}_{in})))) + \mathbf{x}_{in}, \tag{3}$$

where $\mathbf{x}_{in}$ is the feature from the self-attention module. Mix-FFN mixes a $3 \times 3$ convolution and an MLP into each FFN. In our experiments, we will show that a $3 \times 3$ convolution is sufficient to provide positional information for Transformers. In particular, we use depth-wise convolutions for reducing the number of parameters and improving efficiency.

It should be mentioned that CPVT [52] also alleviates this issue by using a $3 \times 3$ Conv to generate conditional PE at different resolutions and then add it to the feature map. Our work conceptually goes one step further as we argue that adding PE to feature map is not necessary in semantic segmentation. Another recent work CvT [56] introduced $3 \times 3$ Convs to model the spatial relationship among tokens. Despite the converging design, our work differs in both motivation and application as we aim to totally remove PEs to handle the training/testing resolution mismatch issue in semantic segmentation. Our intuition started from [67] whereas the same intuition was not discussed in CvT.

### 3.2 Lightweight All-MLP Decoder

SegFormer incorporates a lightweight decoder consisting only of MLP layers and this avoiding the hand-crafted and computationally demanding components typically used in other methods. The key to enabling such a simple decoder is that our hierarchical Transformer encoder has a larger effective receptive field (ERF) than traditional CNN encoders.

The proposed All-MLP decoder consists of four main steps. First, multi-level features $F_i$ from the MiT encoder go through an MLP layer to unify the channel dimension. Then, in a second step, features are up-sampled to 1/4th and concatenated together. Third, a MLP layer is adopted to fuse the concatenated features $F$. Finally, another MLP layer takes the fused feature to predict the segmentation mask $M$ with a $\frac{H}{4} \times \frac{W}{4} \times N_{cls}$ resolution, where $N_{cls}$ is the number of categories. This lets us formulate the decoder as:

$$
\begin{aligned}
\hat{F}_i &= \text{Linear}(C_i, C)(F_i), \forall i \\
\hat{F}_i &= \text{Upsample}(\frac{W}{4} \times \frac{W}{4})(\hat{F}_i), \forall i \\
F &= \text{Linear}(4C, C)(\text{Concat}(\hat{F}_i)), \forall i \\
M &= \text{Linear}(C, N_{cls})(F),
\end{aligned}
\tag{4}
$$

where M refers to the predicted mask, and $\text{Linear}(C_{in}, C_{out})(\cdot)$ refers to a linear layer with $C_{in}$ and $C_{out}$ as input and output vector dimensions respectively.

**Effective Receptive Field Analysis.** For semantic segmentation, maintaining large receptive field to include context information has been a central issue [5, 17, 18]. Here, we use effective receptive field (ERF) [68] as a toolkit to visualize and interpret why our MLP decoder design is so effective on Transformers. In Figure 3, we visualize ERFs of the four encoder stages and the decoder heads for both DeepLabv3+ and SegFormer. We can make the following observations:

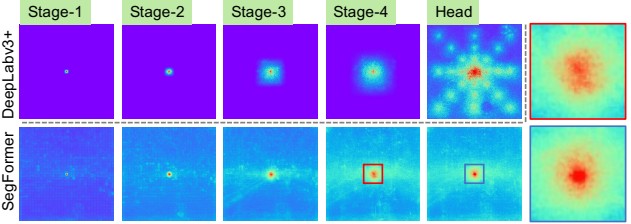

Figure 3: **Effective Receptive Field (ERF) on Cityscapes** (average over 100 images). Top row: Deeplabv3+. Bottom row: Seg-Former. ERFs of the four stages and the decoder heads of both architectures are visualized. Best viewed with zoom in.

- The ERF of DeepLabv3+ is relatively small even at Stage-4, the deepest stage.

- SegFormer's encoder naturally produces local attentions which resemble convolutions at lower stages, while able to output highly non-local attentions that effectively capture contexts at Stage-4.

- As shown with the zoom-in patches in Figure 3, the ERF of the MLP head (blue box) differs from Stage-4 (red box) with a significant stronger local attention besides the non-local attention.

The limited receptive field in CNN requires one to resort to context modules such as ASPP [16] that enlarge the receptive field but inevitably become heavy. Our decoder design benefits from the non-local attention in Transformers and leads to a larger receptive field without being complex. The same decoder design, however, does not work well on CNN backbones since the overall receptive field is upper bounded by the limited one at Stage-4, and we will verify this later in Table 1d,

More importantly, our decoder design essentially takes advantage of a Transformer induced feature that produces both highly local and non-local attention at the same time. By unifying them, our MLP decoder renders complementary and powerful representations by adding few parameters. This is

another key reason that motivated our design. Taking the non-local attention from Stage-4 alone is not enough to produce good results, as will be verified in Table 1d.

### 3.3 Relationship to SETR.

SegFormer contains multiple more efficient and powerful designs compared with SETR [7]:

- We only use ImageNet-1K for pre-training. ViT in SETR is pre-trained on larger ImageNet-22K.

- SegFormer's encoder has a hierarchical architecture, which is smaller than ViT and can capture both high-resolution coarse and low-resolution fine features. In contrast, SETR's ViT encoder can only generate single low-resolution feature map.

- We remove Positional Embedding in encoder, while SETR uses fixed shape Positional Embedding which decreases the accuracy when the resolution at inference differs from the training ones.

- Our MLP decoder is more compact and less computationally demanding than the one in SETR. This leads to a *negligible* computational overhead. In contrast, SETR requires heavy decoders with multiple $3 \times 3$ convolutions.

## 4 Experiments

### 4.1 Experimental Settings

**Datasets:** We used four public datasets: Cityscapes [69], ADE20K [70], and COCO-Stuff [71]. ADE20K is a dataset covering 150 fine-grained semantic concepts consisting of 20210 images. Cityscapes is a driving dataset for semantic segmentation consisting of 5000 fine-annotated high resolution images with 19 categories. COCO-Stuff covers 172 labels and consists of 164k images: 118k for training, 5k for validation, 20k for test-dev and 20k for the test-challenge.

**Implementation details:** We used the *mmsegmentation*[2] codebase and train on a server with 8 Tesla V100. We pre-train the encoder on the Imagenet-1K dataset and randomly initialize the decoder. During training, we applied data augmentation through random resize with ratio 0.5-2.0, random horizontal flipping, and random cropping to $512 \times 512$, $1024 \times 1024$, $512 \times 512$ for ADE20K, Cityscapes and COCO-Stuff. Following [9] we set crop size to $640 \times 640$ on ADE20K for our largest model B5. We trained the models using AdamW optimizer for 160K iterations on ADE20K, Cityscapes, and 80K iterations on COCO-Stuff. Exceptionally, for the ablation studies, we trained the models for 40K iterations. We used a batch size of 16 for ADE20K, COCO-Stuff and a batch size of 8 for Cityscapes. The learning rate was set to an initial value of 0.00006 and then used a "poly" LR schedule with factor 1.0 by default. For simplicity, we *did not* adopt widely-used tricks such as OHEM, auxiliary losses or class balance loss. During evaluation, we rescale the short side of the image to training cropping size and keep the aspect ratio for ADE20K and COCO-Stuff. For Cityscapes, we do inference using sliding window test by cropping $1024 \times 1024$ windows. We report semantic segmentation performance using mean Intersection over Union (mIoU).

### 4.2 Ablation Studies

**Influence of the size of model.** We first analyze the effect of increasing the size of the encoder on the performance and model efficiency. Figure 1 shows the performance vs. model efficiency for ADE20K as a function of the encoder size and, Table 1a summarizes the results for the three datasets. The first thing to observe here is the size of the decoder compared to the encoder. As shown, for the lightweight model, the decoder has only 0.4M parameters. For MiT-B5 encoder, the decoder only takes up to 4% of the total number of parameters in the model. In terms of performance, we can observe that, overall, increasing the size of the encoder yields consistent improvements on all the datasets. Our lightweight model, SegFormer-B0, is compact and efficient while maintaining a competitive performance, showing that our method is very convenient for real-time applications. On the other hand, our SegFormer-B5, the largest model, achieves state-of-the-art results on all three datasets, showing the potential of our Transformer encoder.

---

[2]https://github.com/open-mmlab/mmsegmentation

Table 1: Ablation studies related to model size, encoder and decoder design.

(a) Accuracy, parameters and flops as a function of the model size on the three datasets. "SS" and "MS" means single/multi-scale test.

| Encoder | Params | | ADE20K | | Cityscapes | | COCO-Stuff | |
|---------|--------|--------|---------|--------------|------------|--------------|------------|-------------|
| Model Size | Encoder | Decoder | Flops ↓ | mIoU(SS/MS) ↑ | Flops ↓ | mIoU(SS/MS) ↑ | Flops ↓ | mIoU(SS) ↑ |
| MiT-B0 | 3.4 | 0.4 | 8.4 | 37.4 / 38.0 | 125.5 | 76.2 / 78.1 | 8.4 | 35.6 |
| MiT-B1 | 13.1 | 0.6 | 15.9 | 42.2 / 43.1 | 243.7 | 78.5 / 80.0 | 15.9 | 40.2 |
| MiT-B2 | 24.2 | 3.3 | 62.4 | 46.5 / 47.5 | 717.1 | 81.0 / 82.2 | 62.4 | 44.6 |
| MiT-B3 | 44.0 | 3.3 | 79.0 | 49.4 / 50.0 | 962.9 | 81.7 / 83.3 | 79.0 | 45.5 |
| MiT-B4 | 60.8 | 3.3 | 95.7 | 50.3 / 51.1 | 1240.6 | 82.3 / 83.9 | 95.7 | 46.5 |
| MiT-B5 | 81.4 | 3.3 | 183.3 | 51.0 / 51.8 | 1460.4 | 82.4 / 84.0 | 111.6 | 46.7 |

(b) Accuracy as a function of the MLP dimension $C$ in the decoder on ADE20K.

| $C$ | Flops ↓ | Params ↓ | mIoU ↑ |
|-----|---------|----------|--------|
| 256 | 25.7 | 24.7 | 44.9 |
| 512 | 39.8 | 25.8 | 45.0 |
| 768 | 62.4 | 27.5 | 45.4 |
| 1024 | 93.6 | 29.6 | 45.2 |
| 2048 | 304.4 | 43.4 | 45.6 |

(c) Mix-FFN vs. positional encoding (PE) for different test resolution on Cityscapes.

| Inf Res | Enc Type | mIoU ↑ |
|---------|----------|--------|
| 768×768 | PE | 77.3 |
| 1024×2048 | PE | 74.0 |
| 768×768 | Mix-FFN | 80.5 |
| 1024×2048 | Mix-FFN | 79.8 |

(d) Accuracy on ADE20K of CNN and Transformer encoder with MLP decoder. "S4" means stage-4 feature.

| Encoder | Flops ↓ | Params ↓ | mIoU ↑ |
|---------|---------|----------|--------|
| ResNet50 (S1-4) | 69.2 | 29.0 | 34.7 |
| ResNet101 (S1-4) | 88.7 | 47.9 | 38.7 |
| ResNeXt101 (S1-4) | 127.5 | 86.8 | 39.8 |
| **MiT-B2 (S4)** | 22.3 | 24.7 | 43.1 |
| **MiT-B2 (S1-4)** | 62.4 | 27.7 | 45.4 |
| **MiT-B3 (S1-4)** | 79.0 | 47.3 | **48.6** |

Table 2: **Comparison to state of the art methods on ADE20K and Cityscapes.** SegFormer has significant advantages on #Params (M), #Flops, #Speed and #Accuracy. Note that for SegFormer-B0 we scale the short side of image to {1024, 768, 640, 512} to get speed-accuracy tradeoffs.

| | Method | Encoder | Params ↓ | ADE20K | | | Cityscapes | | |
|--|--------|---------|----------|--------|--------|--------|------------|--------|--------|
| | | | | Flops ↓ | FPS ↑ | mIoU ↑ | Flops ↓ | FPS ↑ | mIoU ↑ |
| Real-Time | FCN [1] | MobileNetV2 | 9.8 | 39.6 | 64.4 | 19.7 | 317.1 | 14.2 | 61.5 |
| | ICNet [11] | - | - | - | - | - | - | 30.3 | 67.7 |
| | PSPNet [15] | MobileNetV2 | 13.7 | 52.9 | 57.7 | 29.6 | 423.4 | 11.2 | 70.2 |
| | DeepLabV3+ [18] | MobileNetV2 | 15.4 | 69.4 | 43.1 | 34.0 | 555.4 | 8.4 | 75.2 |
| | **SegFormer** (Ours) | MiT-B0 | **3.8** | **8.4** | **50.5** | **37.4** | 125.5 | 15.2 | **76.2** |
| | | | | - | - | - | 51.7 | 26.3 | 75.3 |
| | | | | - | - | - | 31.5 | 37.1 | 73.7 |
| | | | | - | - | - | **17.7** | **47.6** | 71.9 |
| Non Real-Time | FCN [1] | ResNet-101 | 68.6 | 275.7 | 14.8 | 41.4 | 2203.3 | 1.2 | 76.6 |
| | EncNet [22] | ResNet-101 | **55.1** | 218.8 | 14.9 | 44.7 | 1748.0 | 1.3 | 76.9 |
| | PSPNet [15] | ResNet-101 | 68.1 | 256.4 | 15.3 | 44.4 | 2048.9 | 1.2 | 78.5 |
| | CCNet [39] | ResNet-101 | 68.9 | 278.4 | 14.1 | 45.2 | 2224.8 | 1.0 | 80.2 |
| | DeeplabV3+ [18] | ResNet-101 | 62.7 | 255.1 | 14.1 | 44.1 | 2032.3 | 1.2 | 80.9 |
| | OCRNet [21] | HRNet-W48 | 70.5 | 164.8 | **17.0** | 45.6 | 1296.8 | **4.2** | 81.1 |
| | GSCNN [33] | WideResNet38 | - | - | - | - | - | - | 80.8 |
| | Axial-DeepLab [72] | AxialResNet-XL | - | - | - | - | 2446.8 | - | 81.1 |
| | Dynamic Routing [73] | Dynamic-L33-PSP | - | - | - | - | **270.0** | - | 80.7 |
| | Auto-Deeplab [48] | NAS-F48-ASPP | - | - | - | 44.0 | 695.0 | - | 80.3 |
| | SETR [7] | ViT-Large | 318.3 | - | 5.4 | 50.2 | - | 0.5 | 82.2 |
| | **SegFormer** (Ours) | MiT-B4 | 64.1 | **95.7** | 15.4 | 51.1 | 1240.6 | 3.0 | 83.8 |
| | **SegFormer** (Ours) | MiT-B5 | 84.7 | 183.3 | 9.8 | **51.8** | 1447.6 | 2.5 | **84.0** |

**Influence of $C$, the MLP decoder channel dimension.** We now analyze the influence of the channel dimension $C$ in the MLP decoder, see Section 3.2. In Table 1b we show performance, flops, and parameters as a function of this dimension. We can observe that setting $C = 256$ provides a very competitive performance and computational cost. The performance increases as $C$ increases; however, it leads to larger and less efficient models. Interestingly, this performance plateaus for channel dimensions wider than 768. Given these results, we choose $C = 256$ for our real-time models SegFormer-B0, B1 and $C = 768$ for the rest.

**Mix-FFN vs. Positional Encoder (PE).** In this experiment, we analyze the effect of removing the positional encoding in the Transformer encoder in favor of using the proposed Mix-FFN. To this end, we train Transformer encoders with a positional encoding (PE) and the proposed Mix-FFN and perform inference on Cityscapes with two different image resolutions: 768×768 using a sliding window, and 1024×2048 using the whole image.

Table 1c shows the results for this experiment. As shown, for a given resolution, our approach using Mix-FFN clearly outperforms using a positional encoding. Moreover, our approach is less sensitive to differences in the test resolution: the accuracy drops 3.3% when using a positional encoding with a

lower resolution. In contrast, when we use the proposed Mix-FFN the performance drop is reduced to only 0.7%. From these results, we can conclude using the proposed Mix-FFN produces better and more robust encoders than those using positional encoding.

**Effective receptive field evaluation.** In Section 3.2, we argued that our MLP decoder benefits from Transformers having a larger effective receptive field compared to other CNN models. To quantify this effect, in this experiment, we compare the performance of our MLP-decoder when used with CNN-based encoders such as ResNet or ResNeXt. As shown in Table 1d, coupling our MLP-decoder with a CNN-based encoder yields a significantly lower accuracy compared to coupling it with the proposed Transformer encoder. Intuitively, as a CNN has a smaller receptive field than the Transformer (see the analysis in Section 3.2), the MLP-decoder is not enough for global reasoning. In contrast, coupling our Transformer encoder with the MLP decoder leads to the best performance. Moreover, for Transformer encoder, it is necessary to combine low-level local features and high-level non-local features instead of only high-level feature.

**Influence of difference encoders.** We select 2 representative Transformer encoders, ViT [6] and Swin [9] and compare with our MiT encoder. As shown in Table 3, with same decoder, *e.g.*MLP decoder, MiT-B2 is 3.1% higher than Swin-T with similar encoder parameters. Moreover, MiT-B5 has much fewer encoder parameters than ViT-large, but is 3+% mIoU higher than ViT-large. These experiments shows our MiT encoder is better than Swin and ViT for semantic segmentataion.

**Influence of difference decoders.** We also test MiT encoder with different decoders. As shown

Table 3: Ablation study of different Transformer encoders and different decoders. All the model are trained on ADE20K with 160K iterations.

| Encoder | Decoder | mIoU | FPS | Decoder GFlops | Decoder Params (M) |
|---|---|---|---|---|---|
| MiT-B2 | UperNet (Swin) | 46.5 | 14.2 | 210.7 | 29.7 |
| MiT-B2 | MLA (SETR) | 46.2 | 9.5 | 87.7 | 4.2 |
| MiT-B2 | MLP (Ours) | 46.5 | 21.4 | 42.1 | 3.3 |
| MiT-B5 | UperNet (Swin) | 50.7 | 5.3 | 210.7 | 29.7 |
| MiT-B5 | MLA (SETR) | 50.9 | 3.8 | 87.7 | 4.2 |
| MiT-B5 | MLP (Ours) | 51.0 | 9.8 | 42.1 | 3.3 |
| Swin-T | MLP (Ours) | 43.4 | 20.6 | 42.8 | 3.6 |
| Swin-T | UperNet (Swin) | 44.5 | 15.4 | 211.3 | 31.4 |
| ViT-L | MLP (Ours) | 47.7 | 4.7 | 0.6 | 0.6 |
| ViT-L | MLA (SETR) | 47.7 | 4.6 | 1.8 | 3.7 |

in Table 3, the mIoUs are similar with different decoders while the proposed MLP decoder has the least parameters and is only $1/8$ of the UperNet decoder in Swin. The MLP decoder is thus an important design towards efficient segmentation.

## 4.3 Comparison to state of the art methods

We now compare our results with existing approaches on the ADE20K [70] and Cityscapes [69]. More experiments about COCO-Stuff [71] are in appendix.

**ADE20K and Cityscapes:** Table 2 summarizes our results including parameters, FLOPS, latency, and accuracy for ADE20K and Cityscapes. In the top part of the table, we report real-time approaches where we include state-of-the-art methods and our results using the MiT-B0 lightweight encoder. In the bottom part, we focus on performance and report the results of our approach and related works using stronger encoders.

On ADE20K, SegFormer-B0 yields 37.4% mIoU using only 3.8M parameters and 8.4G FLOPs, outperforming all other real-time counterparts in terms of parameters, flops, and latency. For instance, compared to DeeplabV3+ (MobileNetV2), SegFormer-B0 is 7.4 FPS, which is faster and keeps 3.4% better mIoU.

Moreover, SegFormer-B5 outperforms all other approaches, including the previous best SETR, and establishes a new state-of-the-art of 51.8%, which is 1.6% mIoU better than SETR while being significantly more efficient.

Table 4: **Comparison to state of the art methods on Cityscapes test set.** IM-1K, IM-22K, Coarse and MV refer to the ImageNet-1K, ImageNet-22K, Cityscapes coarse set and Mapillary Vistas.

| Method | Encoder | Extra Data | mIoU |
|---|---|---|---|
| PSPNet [15] | ResNet-101 | IM-1K | 78.4 |
| PSANet [41] | ResNet-101 | IM-1K | 80.1 |
| CCNet [39] | ResNet-101 | IM-1K | 81.9 |
| OCNet [19] | ResNet-101 | IM-1K | 80.1 |
| Axial-DeepLab [72] | AxiaiResNet-XL | IM-1K | 79.9 |
| SETR [7] | ViT | IM-22K | 81.0 |
| SETR [7] | ViT | IM-22K, Coarse | 81.6 |
| SegFormer | MiT-B5 | IM-1K | 82.2 |
| SegFormer | MiT-B5 | IM-1K, MV | **83.1** |

As also shown in Table 2, our results also hold on Cityscapes. SegFormer-B0 yields 15.2 FPS and 76.2% mIoU (the shorter side of input image being 1024), which represents a 1.3% mIoU improvement and a 2× speedup compared to DeeplabV3+. Moreover, with the shorter side of input image being 512, SegFormer-B0 runs at 47.6 FPS and yields 71.9% mIoU, which is 17.3 FPS faster and 4.2% better than ICNet. SegFormer-B5 archives the best IoU of 84.0%, outperforming all existing

methods by at least 1.8% mIoU, and it runs 5 × faster and 4 × smaller than SETR [7]. On Cityscapes test set, we follow the common setting [18] and merge the validation images to the train set and report results using Imagenet-1K pre-training and also using Mapillary Vistas [74]. As reported in Table 4, using only Cityscapes fine data and Imagenet-1K pre-training, our method achieves 82.2% mIoU outperforming all other methods including SETR, which uses ImageNet-22K pre-training and the additional Cityscapes coarse data. Using Mapillary pre-training, our sets a new state-of-the-art result of 83.1% mIoU.

## 4.4 Robustness to natural corruptions

Model robustness is important for many safety-critical tasks such as autonomous driving [75]. In this experiment, we evaluate the robustness of SegFormer to common corruptions and perturbations. To this end, we follow [75] and generate Cityscapes-C, which expands the Cityscapes validation set with 16 types of algorithmically generated corruptions from noise, blur, weather and digital categories. We compare our method to DeeplabV3+ and other methods as reported in [75]. We also compare with SETR with DeiT Transformer backbone. The results for this experiment are summarized in Table 5.

Our method significantly outperforms previous CNN-based methods, yielding a relative improvement of up to 588% on Gaussian Noise and up to 295% on snow weather. SegFormer also outperforms SETR in general except for one corruption (snow). The results indicate the strong robustness of SegFormer, which we envision to benefit safety-critical applications where robustness is important.

Table 5: **Main results on Cityscapes-C.** "DLv3+", "MBv2", "R" and "X" refer to DeepLabv3+, MobileNetv2, ResNet and Xception. The mIoUs of compared methods are reported from [75].

| Method | Clean | Blur | | | | Noise | | | | Digital | | | | Weather | | | |
|---|---|---|---|---|---|---|---|---|---|---|---|---|---|---|---|---|---|
| | | Motion | Defoc | Glass | Gauss | Gauss | Impul | Shot | Speck | Bright | Contr | Satur | JPEG | Snow | Spatt | Fog | Frost |
| DLv3+ (MBv2) | 72.0 | 53.5 | 49.0 | 45.3 | 49.1 | 6.4 | 7.0 | 6.6 | 16.6 | 51.7 | 46.7 | 32.4 | 27.2 | 13.7 | 38.9 | 47.4 | 17.3 |
| DLv3+ (R50) | 76.6 | 58.5 | 56.6 | 47.2 | 57.7 | 6.5 | 7.2 | 10.0 | 31.1 | 58.2 | 54.7 | 41.3 | 27.4 | 12.0 | 42.0 | 55.9 | 22.8 |
| DLv3+ (R101) | 77.1 | 59.1 | 56.3 | 47.7 | 57.3 | 13.2 | 13.9 | 16.3 | 36.9 | 59.2 | 54.5 | 41.5 | 37.4 | 11.9 | 47.8 | 55.1 | 22.7 |
| DLv3+ (X41) | 77.8 | 61.6 | 54.9 | 51.0 | 54.7 | 17.0 | 17.3 | 21.6 | 43.7 | 63.6 | 56.9 | 51.7 | 38.5 | 18.2 | 46.6 | 57.6 | 20.6 |
| DLv3+ (X65) | 78.4 | 63.9 | 59.1 | 52.8 | 59.2 | 15.0 | 10.6 | 19.8 | 42.4 | 65.9 | 59.1 | 46.1 | 31.4 | 19.3 | 50.7 | 63.6 | 23.8 |
| DLv3+ (X71) | 78.6 | 64.1 | 60.9 | 52.0 | 60.4 | 14.9 | 10.8 | 19.4 | 41.2 | 68.0 | 58.7 | 47.1 | 40.2 | 18.8 | 50.4 | 64.1 | 20.2 |
| ICNet | 65.9 | 45.8 | 44.6 | 47.4 | 44.7 | 8.4 | 8.4 | 10.6 | 27.9 | 41.0 | 33.1 | 27.5 | 34.0 | 6.3 | 30.5 | 27.3 | 11.0 |
| FCN8s | 66.7 | 42.7 | 31.1 | 37.0 | 34.1 | 6.7 | 5.7 | 7.8 | 24.9 | 53.3 | 39.0 | 36.0 | 21.2 | 11.3 | 31.6 | 37.6 | 19.7 |
| DilatedNet | 68.6 | 44.4 | 36.3 | 32.5 | 38.4 | 15.6 | 14.0 | 18.4 | 32.7 | 52.7 | 32.6 | 38.1 | 29.1 | 12.5 | 32.3 | 34.7 | 19.2 |
| ResNet-38 | 77.5 | 54.6 | 45.1 | 43.3 | 47.2 | 13.7 | 16.0 | 18.2 | 38.3 | 60.0 | 50.6 | 46.9 | 14.7 | 13.5 | 45.9 | 52.9 | 22.2 |
| PSPNet | 78.8 | 59.8 | 53.2 | 44.4 | 53.9 | 11.0 | 15.4 | 15.4 | 34.2 | 60.4 | 51.8 | 30.6 | 21.4 | 8.4 | 42.7 | 34.4 | 16.2 |
| GSCNN | 80.9 | 58.9 | 58.4 | 41.9 | 60.1 | 5.5 | 2.6 | 6.8 | 24.7 | 75.9 | 61.9 | 70.7 | 12.0 | 12.4 | 47.3 | 67.9 | 32.6 |
| SETR-DeiT | 78.9 | 64.9 | 65.1 | 59.1 | 65.3 | 54.7 | 60.5 | 51.9 | 69.4 | 74.9 | 69.6 | 74.9 | 58.5 | **44.3** | 64.8 | 68.2 | 39.1 |
| SegFormer-B5 | **82.4** | **69.1** | **68.6** | **64.1** | **69.8** | **57.8** | **63.4** | **52.3** | **72.8** | **81.0** | **77.7** | **80.1** | **58.8** | 40.7 | **68.4** | **78.5** | **49.9** |

# 5 Conclusion

In this paper, we present SegFormer, a simple, clean yet powerful semantic segmentation method which contains a positional-encoding-free, hierarchical Transformer encoder and a lightweight All-MLP decoder. It avoids common complex designs in previous methods, leading to both high efficiency and performance. SegFormer not only achieves new state of the art results on common datasets, but also shows strong zero-shot robustness. We hope our method can serve as a solid baseline for semantic segmentation and motivate further research. One potential limitation is that even our lightest model may still be too heavy for some edge devices. Thus mixed-precision training, pruning, hardware-friendly attention designs and energy consumption are important parts of our future work.

## Broader Impact

Efficiency, accuracy, and robustness are important aspects of AI models. Our work pushes the boundary of semantic segmentation models in these three aspects. We envision that the work will benefit a wide range of safety-critical applications, such as autonomous driving and robot navigation. The proposed method improves the "in-the-wild" robustness of these applications, ultimately leading to better safety. Despite such improvement, we fully understand this work is by no means perfect and there are still many challenges towards reliable real world application. Our models may be subject to biases and other possible undesired mistakes, depending on how they are trained in reality. Our

model may also be used for surveillance similar to other AI recognition methods, even though it is not mainly designed for surveillance applications.

## Acknowledgments and Disclosure of Funding

We thank Ding Liang, Zhe Chen and Yaojun Liu for insightful discussion without which this paper would not be possible. Ping Luo is supported by the General Research Fund of Hong Kong No.27208720.

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
