# SegFormer: Simple and Efficient Design for Semantic Segmentation with Transformers

**Enze Xie**[1] **Wenhai Wang**[2] **Zhiding Yu**[3*] **Anima Anandkumar**[3,4] **Jose M. Alvarez**[3] **Ping Luo**[1]

[1]The University of Hong Kong  [2]Nanjing University  [3]NVIDIA  [4]Caltech
xieenze@hku.hk, wangwenhai362@163.com,
{zhidingy,josea,aanandkumar}@nvidia.com, pluo@cs.hku.hk

## A   Details of MiT Series

In this section, we list some important hyper-parameters of our Mix Transformer (MiT) encoder. By changing these parameters, we can easily scale up our encoder from B0 to B5.

In summary, the hyper-parameters of our MiT are listed as follows:

- $K_i$: the patch size of the overlapping patch embedding in Stage $i$;
- $S_i$: the stride of the overlapping patch embedding in Stage $i$;
- $P_i$: the padding size of the overlapping patch embedding in Stage $i$;
- $C_i$: the channel number of the output of Stage $i$;
- $L_i$: the number of encoder layers in Stage $i$;
- $R_i$: the reduction ratio of the Efficient Self-Attention in Stage $i$;
- $N_i$: the head number of the Efficient Self-Attention in Stage $i$;
- $E_i$: the expansion ratio of the feed-forward layer [1] in Stage $i$;

Table 4 shows the detailed information of our MiT series. Table 3 shows the top-1 accuracy on ImageNet-1K. To facilitate efficient discussion, we assign the code name B0 to B5 for MiT encoder, where B0 is the smallest model designed for real-time, while B5 is the largest model designed for high performance.

## B   More Comparisons

**COCO-Stuff.**   We evaluate SegFormer on the full COCO-Stuff dataset. For comparison, as existing methods do not provide results on this dataset, we reproduce the most representative methods such as DeeplabV3+, OCRNet, and SETR. In this case, the flops on this dataset are the same as those reported for ADE20K. As shown in Table 1, SegFormer-B5 reaches 46.7% mIoU with only 84.7M parameters, which is 0.9% better and 4× smaller than SETR.

Table 1: **Results on COCO-Stuff full dataset** containing all 164K images from COCO 2017 and covers 172 classes.

| Method | Encoder | Params | mIoU |
|---|---|---|---|
| DeeplabV3+ [2] | ResNet50 | **43.7** | 38.4 |
| OCRNet [3] | HRNet-W48 | 70.5 | 42.3 |
| SETR [4] | ViT | 305.7 | 45.8 |
| SegFormer | MiT-B5 | 84.7 | **46.7** |

**Compare with CvT/Swin/PVT.** We implement CvT as an encoder by combining it with our MLP decoder, and keep the other training recipe the same as SegFormer for fair comparison. CvT achieves

---

*Corresponding authors: Zhiding Yu and Ping Luo

35th Conference on Neural Information Processing Systems (NeurIPS 2021).

| Encoder | Decoder | mIoU | ImageNet Top-1 | Encoder Params (M) |
|---------|---------|------|----------------|--------------------|
| MiT-B2 (Ours) | MLP (Ours) | 46.5 | 81.4 | 24.2 |
| CvT-13 | MLP (Ours) | 43.7 | 81.6 | 20.0 |
| Swin-T | UperNet | 44.5 | 81.2 | 28.0 |
| PVT-S | SemFPN | 43.2 | 79.8 | 24.5 |

Table 2: Compare with CvT/Swin/PVT on ADE20K segmentation and ImageNet-1K classification.

| Method | GFLOPs | Params (M) | ImageNet Top 1 |
|--------|--------|------------|----------------|
| MiT-B0 | 0.6 | 3.4 | 70.3 |
| MiT-B1 | 2.1 | 13.1 | 78.5 |
| MiT-B2 | 4.0 | 24.2 | 81.4 |
| MiT-B3 | 6.9 | 44.0 | 83.0 |
| MiT-B4 | 10.1 | 60.8 | 83.4 |
| MiT-B5 | 11.8 | 81.4 | 83.7 |

Table 3: Top-1 Accuracy on ImageNet-1K from MiT-B0 to MiT-B5.

good performance but is lower than SegFormer. We hypothesize that the decreased performance is caused by the fact that CvT was initially designed to solve image classification tasks. Table 2 shows the single scale segmentation performance and the ImageNet-1K Top-1 accuracy of the corresponding backbones. Note that all encoders here are pre-trained on ImageNet-1K with shape 224x224. The results of Swin Transformer on semantic segmentation are reported from its paper. The results of CvT, Swin and PVT-S on ImageNet-1K are also reported from their papers.

## C  More Qualitative Results on Mask Predictions

Figure 1 shows qualitative results on Cityscapes, where SegFormer provides better details than SETR and smoother predictions than DeeplabV3+. In Figure 2, we present more qualitative results on Cityscapes, ADE20K and COCO-Stuff, compared with SETR and DeepLabV3+.

Compared to SETR, our SegFormer predicts masks with significantly finer details near object boundaries because our Transformer encoder can capture much higher resolution features than SETR, which preserves more detailed texture information. Compared to DeepLabV3+, SegFormer reduces long-range errors benefit from the larger effective receptive field of Transformer encoder than ConvNet.

## D  More Visualization on Effective Receptive Field

In Figure 3, we select some representative images and effective receptive field (ERF) of DeepLabV3+ and SegFormer. Beyond larger ERF, the ERF of SegFormer is more sensitive to the context of the image. We see SegFormer's ERF learned the pattern of roads, cars, and buildings, while DeepLabV3+'s ERF shows a relatively fixed pattern. The results also indicate that our Transformer encoder has a stronger feature extraction ability than ConvNets.

## E  More Comparison of DeeplabV3+ and SegFormer on Cityscapes-C

In this section, we detailed show the zero-shot robustness compared with SegFormer and DeepLabV3+. Following [5], we test 3 severities for 4 kinds of "Noise" and 5 severities for the rest 12 kinds of corruptions and perturbations.

As shown in Figure 4, with severity increase, DeepLabV3+ shows a considerable performance degradation. In contrast, the performance of SegFormer is relatively stable. Moreover, SegFormer has significant advantages over DeepLabV3+ on all corruptions/perturbations and all severities, demonstrating excellent zero-shot robustness.

| | Output Size | Layer Name | Mix Transformer | | | | | |
|---|---|---|---|---|---|---|---|---|
| | | | B0 | B1 | B2 | B3 | B4 | B5 |
| **Stage 1** | $\frac{H}{4} \times \frac{W}{4}$ | Overlapping Patch Embedding | $K_1 = 7;\ S_1 = 4;\ P_1 = 3$ | | | | | |
| | | | $C_1 = 32$ | $C_1 = 64$ | | | | |
| | | Transformer Encoder | $R_1 = 8$ $N_1 = 1$ $E_1 = 8$ $L_1 = 2$ | $R_1 = 8$ $N_1 = 1$ $E_1 = 8$ $L_1 = 2$ | $R_1 = 8$ $N_1 = 1$ $E_1 = 8$ $L_1 = 3$ | $R_1 = 8$ $N_1 = 1$ $E_1 = 8$ $L_1 = 3$ | $R_1 = 8$ $N_1 = 1$ $E_1 = 8$ $L_1 = 3$ | $R_1 = 8$ $N_1 = 1$ $E_1 = 4$ $L_1 = 3$ |
| **Stage 2** | $\frac{H}{8} \times \frac{W}{8}$ | Overlapping Patch Embedding | $K_2 = 3;\ S_2 = 2;\ P_2 = 1$ | | | | | |
| | | | $C_2 = 64$ | $C_2 = 128$ | | | | |
| | | Transformer Encoder | $R_2 = 4$ $N_2 = 2$ $E_2 = 8$ $L_2 = 2$ | $R_2 = 4$ $N_2 = 2$ $E_2 = 8$ $L_2 = 2$ | $R_2 = 4$ $N_2 = 2$ $E_2 = 8$ $L_2 = 3$ | $R_2 = 4$ $N_2 = 2$ $E_2 = 8$ $L_2 = 3$ | $R_2 = 4$ $N_2 = 2$ $E_2 = 8$ $L_2 = 8$ | $R_2 = 4$ $N_2 = 2$ $E_2 = 4$ $L_2 = 6$ |
| **Stage 3** | $\frac{H}{16} \times \frac{W}{16}$ | Overlapping Patch Embedding | $K_3 = 3;\ S_3 = 2;\ P_3 = 1$ | | | | | |
| | | | $C_3 = 160$ | $C_3 = 320$ | | | | |
| | | Transformer Encoder | $R_3 = 2$ $N_3 = 5$ $E_3 = 4$ $L_3 = 2$ | $R_3 = 2$ $N_3 = 5$ $E_3 = 4$ $L_3 = 2$ | $R_3 = 2$ $N_3 = 5$ $E_3 = 4$ $L_3 = 6$ | $R_3 = 2$ $N_3 = 5$ $E_3 = 4$ $L_3 = 18$ | $R_3 = 2$ $N_3 = 5$ $E_3 = 4$ $L_3 = 27$ | $R_3 = 2$ $N_3 = 5$ $E_3 = 4$ $L_3 = 40$ |
| **Stage 4** | $\frac{H}{32} \times \frac{W}{32}$ | Overlapping Patch Embedding | $K_4 = 3;\ S_4 = 2;\ P_4 = 1$ | | | | | |
| | | | $C_4 = 256$ | $C_4 = 512$ | | | | |
| | | Transformer Encoder | $R_4 = 1$ $N_4 = 8$ $E_4 = 4$ $L_4 = 2$ | $R_4 = 1$ $N_4 = 8$ $E_4 = 4$ $L_4 = 2$ | $R_4 = 1$ $N_4 = 8$ $E_4 = 4$ $L_4 = 3$ | $R_4 = 1$ $N_4 = 8$ $E_4 = 4$ $L_4 = 3$ | $R_4 = 1$ $N_4 = 8$ $E_4 = 4$ $L_4 = 3$ | $R_4 = 1$ $N_4 = 8$ $E_4 = 4$ $L_4 = 3$ |

Table 4: **Detailed settings of MiT series.** Our design follows the principles of ResNet [6]. (1) the channel dimension increase while the spatial resolution shrink with the layer goes deeper. (2) Stage 3 is assigned to most of the computation cost.

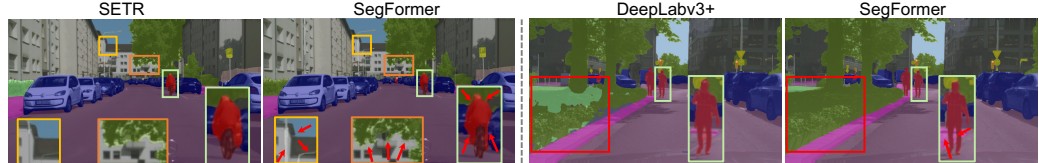

Figure 1: **Qualitative results on Cityscapes.** Compared to SETR, our SegFormer predicts masks with substantially finer details near object boundaries. Compared to DeeplabV3+, SegFormer reduces long-range errors as highlighted in red. Best viewed in screen.

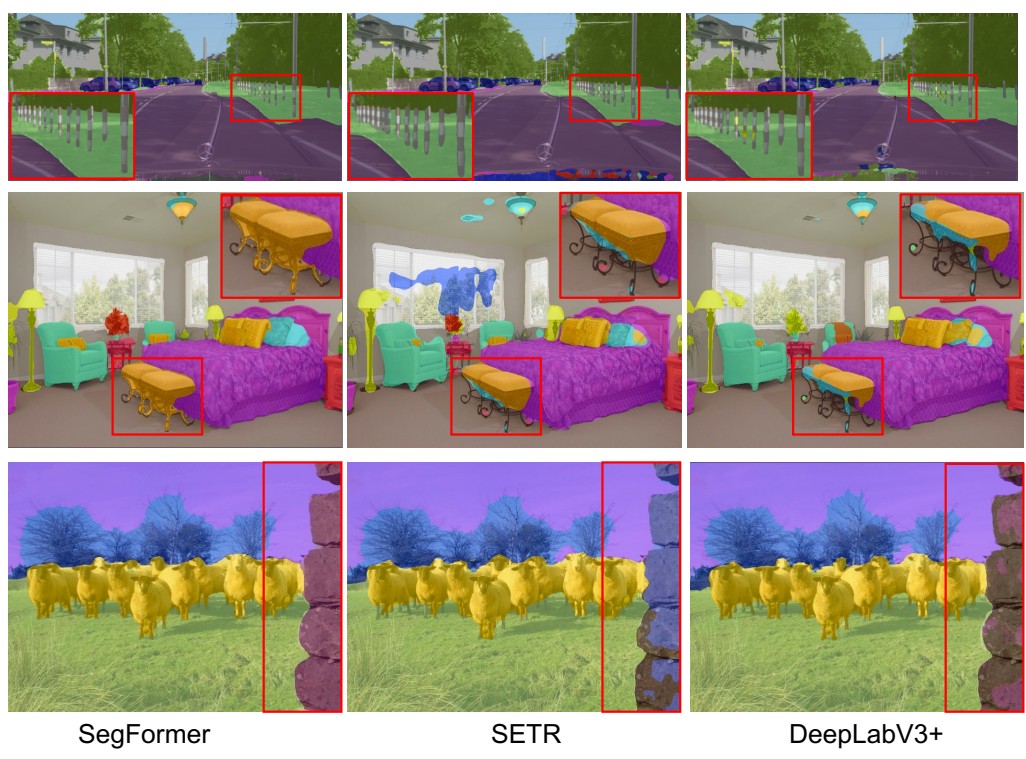

SegFormer                          SETR                      DeepLabV3+

Figure 2: **Qualitative results on Cityscapes, ADE20K and COCO-Stuff.** First row: Cityscapes. Second row: ADE20K. Third row: COCO-Stuff. Zoom in for best view.

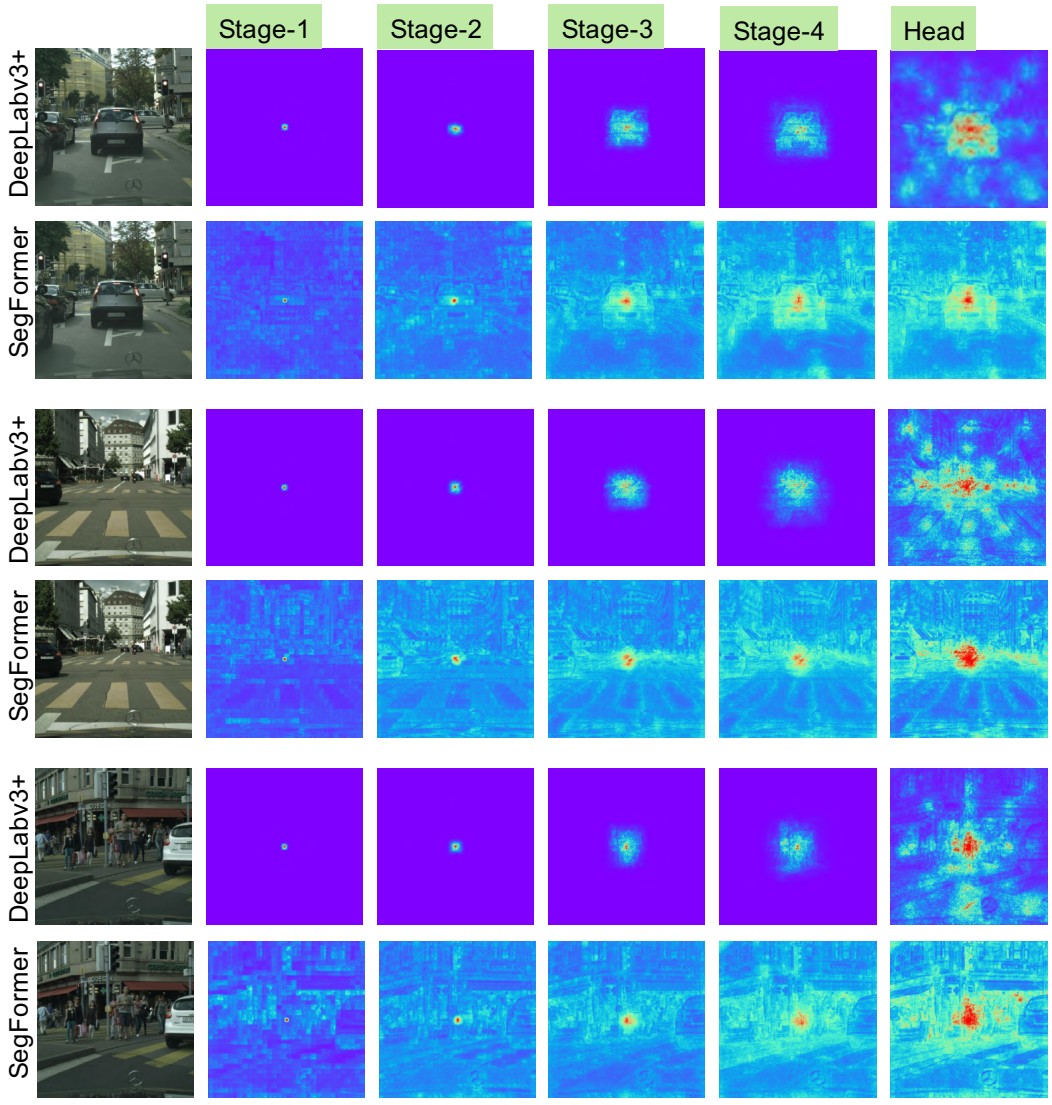

Figure 3: **Effective Receptive Field on Cityscapes.** ERFs of the four stages and the decoder heads of both architectures are visualized.

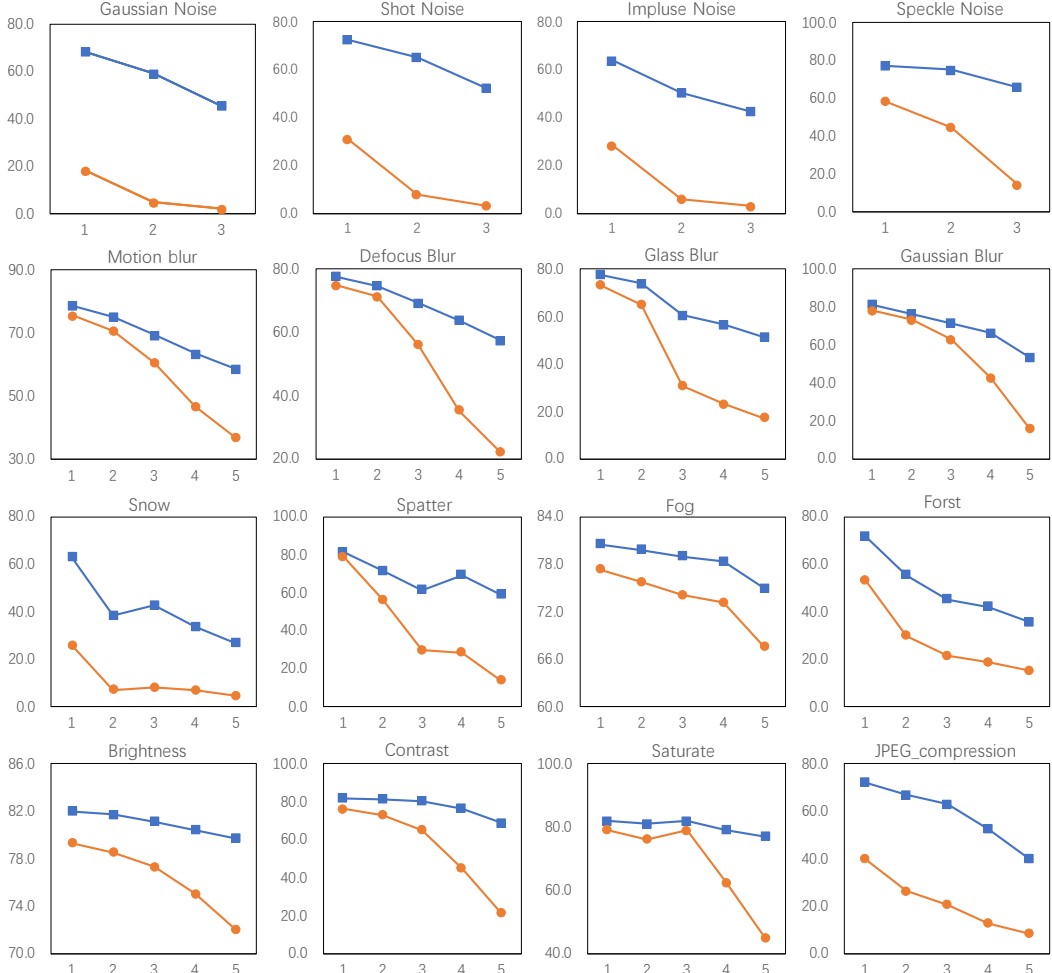

Figure 4: **Comparison of zero shot robustness on Cityscapes-C between SegFormer and DeepLabV3+.** Blue line is SegFormer and orange line is DeepLabV3+. X-Axis means corrupt severity and Y-Axis is mIoU. Following[5], we test 3 severities for "Noise" and 5 severities for the rest.