# OpenReview forum: "SegFormer: Simple and Efficient Design for Semantic Segmentation with Transformers"
_NeurIPS.cc/2021/Conference — NeurIPS 2021 Poster_

### Official Review · Reviewer_n5uX · 2021-07-03

**Rating:** 5
**Confidence:** 5

**Summary:**

This paper introduces SegFormer for semantic segmentation. The SegFormer consists:
(1) A Transformer encoder that extracts multi-scale features. The authors claim their Transformer encoder to be novel in the sense that it is "positional-encoding-free" and "hierarchical".
(2) A "MLP" based decoder that aggregates information from different layers. Although the authors claim it to be "MLP", the decoder is essentially a stack of several 1x1 convolutions with bilinear interpolation to upsample low-resolution features.

The authors further explore scaling the Transformer encoder which ends up with a series of models with different number of parameters and FLOPs. The authors show their models are competitive with SOTA on multiple semantic segmentation datasets and claim that their model is more robust on corrupted images than DeepLabV3+ (on Cityscapes-C).

**Limitations And Societal Impact:**

yes

**Main Review:**

## Limitations of the paper

First of all, I don't agree with the position of this paper. The authors claim that both the Transformer encoder and the new "MLP" decoder are equally important to make SegFormer work. However, I don't see any reason why other decoders cannot be used with the SegFormer Transformer encoder, and why other Transformer encoder cannot be used with Segformer decoder. For me, this paper is more like a study of Transformer design that is not making a proper comparison with other Transformers: the authors did not show their number on ImageNet classification, which is fine if the model only works well for semantic segmentation. However, the comparison with other Transformers on semantic segmentation is completely unfair which I will explain later.

Although this paper shows good performance on multiple semantic segmentation datasets, it has two major problems: (A) the authors overclaimed their contribution (or did not clearly summarize their contribution) and (B) the experiments in this paper cannot support claims made by the authors. Next, I will explain my reasons.

### A. The authors overclaimed their contribution (or did not clearly summarize their contribution)

1. The authors claim they propose a "novel positional-encoding-free and hierarchical Transformer encoder" (L61).
    * In L165, it looks like the authors follow CPVT [54] by using 3x3 convolution to replace positional encoding. Furthermore, in L166-167, the authors argue "positional encoding is actually not necessary for semantic segmentation" without any explanation. I'm not sure how the authors come up with this conclusion, since it is something not obvious. In the Table 1 (c), the authors only compare "PE" with "Mix-FFN", but it is not enough to say positional encoding is not necessary. Given these facts, I don't think "positional-encoding-free" or "Mix-FFN" is something contributed by the authors, as there are many works already uncover this fact like CPVT [54] and CvT [58].

    * As for the hierarchical Transformer, I find it is very hard to understand what is the key difference with other hierarchical Transformers like PvT [8] from current description (the current text only describes "Hierarchical Feature Representation", "Overlapped Patch Merging", "Efficient Self-Attention" which seems to be already used by multiple hierarchical Transformer papers).

2. The authors claim their "All-MLP" decoder is lightweight and less complex than other decoder. First of all, "complex" is very subjective. Why one decoder is more complex than another decoder without a proper metric? Second, the decoder used in the paper is **not** "All-MLP": the decoder is essentially several 1x1 convolutions. And here is the question again: why 1x1 convolutions are less "complex" than 3x3 convolutions?

### B. The experiments in this paper cannot support claims made by the authors

1. The experiments do not show both proposed components (new Transformer encoder and "MLP" decoder) are necessary.
    * In Table 1, the authors only compares their Transformers with CNN backbones which is not a very fair comparison. In order to really show the benefit of this new Transformer, the authors should compare their Transformer with other Transformer designs (e.g., PvT, CvT, Swin Transformer etc.) to really show this is a better Transformer.
    * As for the decoder design, the authors only compare their decoder with different channel dimensions. However, the claim made by the authors is the "MLP" decoder is better than other "more complex" decoder. In order to backup this claim, the authors should compare the "MLP" decoder with the "more complex" decoder they refer to.

2. Design of the experiments is not consistent with the position of the paper. The authors position their paper as a Transformer design for semantic segmentation. Thus, it is okay to not comparing with other Transformer models on ImageNet. However, the authors show at least have a proper comparison with existing Transformers to show why their design is better than other Transformers. For example, the authors can compare their Transformer with Swin Transformer using **the same decoder and the same training recipe**. However, from the description of the paper, I do not see why this special design is more suitable than other Transformer specifically for semantic segmentation.

## Summary

In summary: (1) I believe the claims made by the authors in this paper is not strong enough, (2) experiments cannot support the claims made by the authors as well; and (3) the position of this paper is weird. I'm especially concerned about (2) and (3). I do not see why the design of this Transformer is specifically suitable for semantic segmentation, if the model does not perform better in ImageNet classification. Since the paper never offers an apple-to-apple comparison with other Transformer models for semantic segmentation task, I'm afraid the authors might have tuned training parameters very hard to make it work on semantic segmentation (while other Transformers mainly focus on reporting numbers for classification). If this is the case, I don't feel this paper is very valuable and I think it is a clear rejection. But I hope to see authors explanation if I make any mistake.



--------

\#### post-rebuttal update ####

I have read carefully all reviews and author response. The authors have addressed most concerns well and the major disagreement between reviewers (e.g., me and Reviewer GD12) and the authors lies in the position of this paper. I have to say that I'm still not fully convinced and I believe it is better to position the paper as a new ViT design instead of a specific model design for semantic segmentation. However, the authors argue they mainly show improvements on semantic segmentation which is the reason that they position it as a model for semantic segmentation specifically, which is a fair point and I respect their decision.

Besides the problem of "novelty" raised by all other reviewers, another major problem in the original submission is that there is no fair comparison which leads to my original decision of rejection. The authors have addressed this problem by running the suggested experiments and thus I will raise my score to borderline. However, I believe the paper requires a major revision to take into account all these discussions and I'm not sure if the authors will make the revision as they promised since I cannot see the revised version. So I will change my score to 5 and let the AC to decide whether this should be taken into account.


**Time Spent Reviewing:**

2 hours

---

> ### Author Response · Authors · 2021-08-10
> **Rebuttal by Paper1675 Authors**
>
> Dear Reviewer n5uX,
> Thank you for the detailed review. We will address your concerns below.
>
> **General Response:**
> The main comment is “why other decoders cannot be used with the SegFormer Transformer encoder, and why other Transformer encoders cannot be used with the Segformer decoder”. We believe this is caused by a misinterpretation of the message we are trying to convey in this work. Below we provide some clarifications.
>
> Contrary to reviewer's perception, the connotation of this work is that we assume readers would by default agree that using other (heavier) decoders *is certainly possible, but not necessary*. Indeed, there is no reason why a more heavily parameterized but reasonably designed decoder would hurt the performance. However, our goal is to find simpler and lightweight designs that maintain good performance, while promoting efficiency. At the same time, we want to provide insights on why it works.
>
> In this regard, we disagree on the conclusion on the position of this paper, although we agree that additional ablation studies are needed for a better understanding. To this end, we have conducted additional studies on different combinations of encoder/decoder architectures and listed in **New experiments with different encoder/decoder combinations** in the **Author response to all reviewers**. As demonstrated in these ablation studies, the proposed simple decoder design indeed helps to promote efficiency while maintaining good performance. Below, we address the other comments.
>
>
>
> **Q1: Report ImageNet classification result.**
> A1: Thanks for the suggestion. We actually have reported the results in Table 2 of Appendix. We will move it to the main paper to make it clearer.
>
> | method 	| GFLOPs 	| Params (M) 	| Top 1 	|
> |--------	|:------:	|:----------:	|:-----:	|
> | MiT-B0 	|   0.6  	|     3.4    	|  70.3 	|
> | MiT-B1 	|   2.1  	|    13.1    	|  78.5 	|
> | MiT-B2 	|   4.0  	|    24.2    	|  81.4 	|
> | MiT-B3 	|   6.9  	|    44.0    	|  83.0 	|
> | MiT-B4 	|  10.1  	|    60.8    	|  83.4 	|
> | MiT-B5 	|  11.8  	|    81.4    	|  83.7 	|
>
> **Q2: Relationship of Mix-FFN with CPVT/CvT.**
> A2: SegFormer has several key differences compared with CPVT and CvT. CPVT actually does not remove the concept of positional encoding (PE). The way in which CPVT uses conditional Positional Encodings (CPE) is the same as ViT. It adds the CPE to the feature map. It only provides an alternative way to use 3x3 conv to generate the PE. In this work, we find that adding PE to the feature map is not necessary, while directly using 3x3 conv is enough. Conceptually, our work goes one step further than CVPT in terms of removing PE.
>
> Our design does show certain similarities to CvT. However, CvT is mainly designed as an image classification backbone, and the introduction of conv in CvT was proposed to model the spatial relationship between tokens. In our case, the conv design is mainly motivated to eliminate PEs to address the training/testing resolution mismatch issues in segmentation problems. We feel that this is a converging design with completely different motivations. Indeed, while we started from [a] which indicates 3x3 conv can leak positional information, the same motivation and paper are not discussed nor cited in CvT.
>
> More importantly, we emphasize that CvT is a **concurrent work** released to arXiv (03/29/2021). We do not feel that it is  an example that weakens the technical contribution from this work.
>
> [a] Islam et al., How much position information do convolutional neural networks encode, ICLR2020
>
> **Q3: Key difference of encoder between PVT and this paper.**
> A3: We have clarified this in our general response to all reviewers. Please kindly check our answer in **Difference between SegFormer and PVTv1**. We feel that a potential misunderstanding here is that designs from PVTv1 are indeed necessary to achieve high efficiency, but are not enough to guarantee the SOTA efficiency-performance pareto-frontier of SegFormer.
>
> This is clearly verified in our ablation studies with different combinations of encoders/decoders, as well as Fig. 1 in the main paper. The large improvement indicates that several novel designs not presented in PVTv1, including:
> 1) The positional embedding free design
> 2) The simple MLP decoder design
> 3) Overlapped patch embedding
>
> , which can significantly benefit segmentation problems. The conclusion “... swapping out an image classification with a pixel classification target is of limited novelty” therefore fails to correctly capture this work.
>
> **Q4: Why is MLP decoder less complex than other decoders?**
> A4: Here, by saying “complex”, we refer to the computation complexity, model size and speed. Previous decoders such as ASPP and UperNet often contain dilated 3x3 convs, pyramid poolings, and multiple feature fusion steps, which often leads to larger FLOPs. Instead, our MLP decoder does not have 3x3 convs, which is simple and efficient.
>
> We report the result and decoder FLOPs and parameters as below:
>
> | Encoder 	| Decoder        	| mIoU 	|  FPS 	| Decoder  Flops	| Decoder Params (M) 	|
> |---------	|----------------	|:----:	|:----:	|:---------------:	|:---------------:	|
> | MiT-B2  	| UperNet (Swin) 	| 45.3 	| 14.2 	|      210.7      	|       29.7      	|
> | MiT-B2  	| MLA (SETR)     	| 45.2 	|  9.5 	|       87.7      	|       4.2       	|
> | MiT-B2  	| MLP (Ours)     	| 45.4 	| 21.4 	|       42.1      	|       3.3       	|
>
>
> **Q5: Why is 1x1 conv less complex than 3x3 conv?**
> A5: This is because 1x1 convs only have 1/9=11% parameters and FLOPs compared to 3x3 convs. We will make it clearer in the revised version of the paper.
>
> **Q6: Compare SegFormer with PVT/CvT/Swin in Table1.**
> A6: We have partly addressed this in our answer to **Missing comparison to Swin Transformers and PVT** in the **Author Response to All Reviewers**. As mentioned, we consider CvT and Swin as concurrent work. For both PVT and Swin Transformer, we have compared with them in Fig. 1 of the main paper.
>
> We also implement CvT as an encoder by combining it with our MLP decoder, and keep the other training recipe the same as SegFormer for fair comparison. The method achieves good performance but is lower than SegFormer. We hypothesize that the decreased performance is caused by the fact that CvT was initially designed to solve image classification tasks. We will update it in future versions.
> The following table shows the single scale segmentation performance and the ImageNet-1K Top-1 accuracy of the corresponding backbones. Note that all encoders here are pre-trained on ImageNet-1K with shape 224x224. The results of Swin Transformer on semantic segmentation are reported from its paper. The results of CvT, Swin and PVT-S on ImageNet-1K are also reported from their papers. One could see that the image classification performance of the SegFormer backbone is comparable to Swin and CvT.
>
> | Encoder 	| Decoder    	| Training Iterations 	| mIoU 	| ImageNet Top-1  	| Encoder Params (M) 	|
> |---------	|------------	|:-------------------:	|:----:	|:-------------:	|:---------------:	|
> | MiT-B2  	| MLP (Ours) 	|         160K        	| 46.5 	|      81.4     	|       24.2      	|
> | CvT-13  	| MLP (Ours) 	|         160K        	| 43.7 	|      81.6     	|       20.0      	|
> | Swin-T  	| UperNet    	|         160K        	| 44.5 	|      81.2     	|       28.0      	|
> | PVT-S   	| SemFPN     	|         160K        	| 43.2 	|      79.8     	|       24.5      	|
>
>
> For other different combinations of encoder/decoder architectures, the reviewer may refer to the complete ablation studies in our answer to **New experiments with different encoder/decoder combinations** in the **Author response to all reviewers**.
>
> **Q7: Compare MLP decoder with other more complex decoders.**
> A7: Thanks for the suggestion. We compare the proposed MLP decoder with several decoders. One can see that the performance is similar but our MLP decoder has the best efficiency, with the smallest FLOPs, smallest number of parameters, and fastest FPS.
>
> | Encoder 	| Decoder        	| mIoU 	|  FPS 	| Decoder GFLOPs	| Decoder Params (M) 	|
> |---------	|----------------	|:----:	|:----:	|:---------------:	|:---------------:	|
> | MiT-B2  	| UperNet (Swin) 	| 45.3 	| 14.2 	|      210.7      	|       29.7      	|
> | MiT-B2  	| MLA (SETR)     	| 45.2 	|  9.5 	|       87.7      	|       4.2       	|
> | MiT-B2  	| MLP (Ours)     	| 45.4 	| 21.4 	|       42.1      	|       3.3       	|
>
> **Q8: Concern about heavily tuned hyper-parameters.**
> A8: We are sorry, but we think there is a significant misunderstanding here. Our source code is available in the supplementary. All the hyper-parameters are following common settings in MMSegmentation, including data loading, augmentation, training iterations, learning rate etc. For example, the data augmentation is using the default settings in MMSegmentation. The learning rate and training iterations are set the same as Swin Transformer. We have also reported the number of ImageNet classification in our appendix, which is comparable to Swin and PVT on ImageNet-1K training. It is clear that we did not heavily overtune hyper-parameters on segmentation tasks. We kindly ask the reviewer to check our paper and the code for better understanding.

---

> > ### Comment · Reviewer_n5uX · 2021-08-17
> > **Response to authors rebuttal**
> >
> > I thank the authors for their detailed rebuttal, unfortunately, I am still not fully convinced by the response from the authors, especially the position of this paper. Please see my explanation below:
> >
> > _**1. regarding the positioning of this paper**_
> >
> > I particularly have **two concerns** on the positioning of this paper that are not well-addressed by the authors.
> >
> > The first one is that I don't see why the proposed MiT model is specific for semantic segmentation. SETR already shows the effectiveness of Transformer-based backbones for semantic segmentation, so messages like "Transformer models are better suited" is not surprising and new. Besides, from the ImageNet classification results, it looks like the improvement on semantic segmentation comes from the improvement on image classification. For example, the authors compare MiT-B2 with PVT-S, the MiT-B2 has 81.4% top-1 accuracy whereas PVT-S only has 79.8% top-1 accuracy. Thus, using MiT-B2 as backbone has better performance on semantic segmentation is not surprising. A analogy with CNN backbone is like replacing ResNet with ResNeXt and observe improvements on downstream tasks. Thus, there is no evidence showing any of the design of MiT is specific to semantic segmentation.
> >
> > The second one is on the decoder design side. If the authors believe current heavy decoder design is unnecessary, why not focus on the decoder design itself? Why convolute the decoder design with the backbone design? From the writing of this paper, it feels like the light-weight decoder is only possible together with the use of MiT backbone.
> >
> > In summary, I feel that the technical contribution of backbone design part _itself_ is not enough **under the context of vision transformer community** (as other reviewers also point out that many design choices are used by concurrent works that appear on ArXiv earlier this year) and the technical contribution of decoder design part _itself_ is not enough **under the context of semantic segmentation community** (as I point out, at least there is no thorough study on this).
> >
> > I feel the authors do not understand my concerns correctly according to the response and my suggestion is, it seems like the position of this paper is more on vision transformer design side, and the authors should focus on it and convince the experts on vision transformer that the contribution of this paper is solid (i.e., the proposed component is important and of value to the community). If the authors are still interested in semantic segmentation and want to include the vision transformer design part, then I'm unfortunately not convinced by the story if the authors cannot provide more evidence on why such design is specifically good for semantic segmentation.
> >
> > _**2. regarding the difference with CvT and PvT**_
> >
> > In the response, the authors try to argue the main difference is CvT and PvT are designed for image classification ("CvT is mainly designed as an image classification backbone, and the introduction of conv in CvT was proposed to model the spatial relationship between tokens" and "The large improvement indicates that several novel designs not presented in PVTv1, including: The simple MLP decoder design"). These arguments do not hold if we only consider the backbone design itself.
> >
> > _**3. regarding the new experiment in the general response**_
> >
> > I feel 40k training schedule is not convincing. Models usually do not converge at 40k iterations on ADE20K datasets and some models may converge faster than other.

---

> > > ### Author Response · Authors · 2021-08-19
> > > **Rebuttal by Paper1675 Authors**
> > >
> > > **A1: Position of the paper**
> > >
> > > We thank the reviewer for following up, but disagree with the position the reviewer is taking to criticize this work. We are concerned with the subjective and policing messages in some parts of the review.
> > >
> > >
> > > **Self-contradicting comments**
> > >
> > > In the original review, the reviewer casted doubts by saying “I do not see why the design of this Transformer is specifically suitable for semantic segmentation, if the model does not perform better in ImageNet classification”. In the follow-up review, however, the point suddenly changed to “from the ImageNet classification results, it looks like the improvement on semantic segmentation comes from the improvement on image classification ... Thus, using MiT-B2 as backbone has better performance on semantic segmentation is not surprising”. The above comments are self-contradicting and we are not clear as to what results would really convince the reviewer. Does the reviewer hope to see stronger results on ImageNet classification, or weaker ones?
> > >
> > >
> > > **Unreasonably picky requirements**
> > >
> > > The reviewer seems to have an extraordinary “high standard”. This is the first time a stronger image classification result is becoming a sin for rejecting. Our work follows the predominant stream of semantic segmentation methods where there do exist a positive correlation between backbone performance on ImageNet and the downstream semantic segmentation task. A good performance on image classification should not become a reason to argue that the design is not well designed for semantic segmentation.
> > >
> > > We also do not understand why purely focusing on one module (encoder or decoder) is the single universal golden standard of principled research, given that **the context of semantic segmentation** is highly emphasized by the reviewer. Both encoder and decoder are important components of semantic segmentation problems. We have shown that the simplicity of the MLP decoder is important to achieve the SOTA accuracy-efficiency performance. We also provide an explanatory framework showing that complementary representation with both local details and global context emerges under this design, which is obviously meaningful within the context of segmentation. We consider the fact that this MLP decoder design is specific to the transformer encoder an important contribution. Because this exactly indicates that our design is not a simple A+B combination, but is well-motivated and highly coupled towards addressing segmentation problems. We do not understand why the reviewer insists on separating them and criticizes it as “convoluted”.
> > >
> > >
> > > **Selective/twisted interpretation and double standard**
> > >
> > > The reviewer seems to be selective in acknowledging the contributions of this work. Compared with SETR, SegFormer has much fewer parameters/FLOPs, faster speed and higher accuracy, which are all important considerations in semantic segmentation. The positional encoding-free design is clearly an important design to address segmentation-related issues, as shown in our experiment. Following the reviewer’s logic, if these were not considered important issues in semantic segmentation, then papers like ICNet and DeepLabv1/v2/v3+ should also not be accepted because their scopes and contributions are quite similar to this work at high level.
> > >
> > > The reviewer mentioned “A analogy with CNN backbone is like replacing ResNet with ResNeXt and observe improvements on downstream tasks”. This is a completely false analogy since it:
> > > 1) Incorrectly implies that our method is not novel, just like ResNeXt is a proposed, well-known architecture.
> > > 2) Selectively ignores several key designs (such as the positional-encoding-free design) that are shown to clearly benefit segmentation problems.
> > > 3) Falsely exaggerated the gains in our backbone. Our ImageNet classification performance is at a similar level with Swin and CvT, and our good accuracy-efficiency performance in segmentation is clearly a result of the segmentation-friendly designs.
> > >
> > > The reviewer mentioned “the technical contribution of decoder design part itself is not enough under the context of semantic segmentation community (as I point out, at least there is no thorough study on this)”. We have clearly shown in the paper and in the additional experiments that:
> > > 1) The proposed decoder works well with other type of encoders.
> > > 2) The proposed decoder reduces considerable FLOPs and computation costs, while maintaining very good performance.
> > > 3) The proposed decoder is well-motivated (local+global complementary representation) by leveraging the emerging properties in Transformers.
> > >
> > > Based on the reviewer's logic, Mask R-CNN should have also not been accepted by ICCV and even won the best paper because it’s simply adding a mask head and RoI Align on top of Faster R-CNN. We believe simple is best and should not be the original sin for rejecting papers. On the contrary, it should be encouraged as long as it is well-motivated and makes sense.
> > >
> > > As authors, we are always very open to constructive criticism. But we sincerely hope that the reviewer avoids criticizing the paper with a predetermined position or just for the sake of arguing for rejection.
> > >
> > >
> > > **A2: Difference with CvT and PVT**
> > >
> > > We kindly ask the reviewer please do not ignore the fact that CvT is a concurrent work. We believe it is **against the conference policy** to use concurrent work as an argument for weakened contribution, especially when there is clear evidence showing that these are converging techniques with completely different motivations and applications.
> > >
> > > We did not argue that the difference with PVTv1 is designed for image classification. PVTv1 is a general purpose backbone which does include semantic segmentation beyond image classification. Please do not twist our previous rebuttal. We have clearly listed the difference between SegFormer and PVTv1 and the resulting huge difference in performance. As a result, we don’t understand the detailed logic that supports “these arguments do not hold if we only consider the backbone design itself”.
> > >
> > > **A3: Concern about new experiment with 40k iteration training**
> > >
> > > We want to emphasize that the conclusion on 40K iteration training is showing a converging trend with that on 160K. Several reasons:
> > > 1) Several previous works such as PVT use 40k iteration as the default training setting.
> > > 2) SegFormer-40K is only 1% lower than SegFormer-160K, which shows that the model converges well with 40K iterations.
> > > 3) We have shown that with both 40K and 160K iterations training, SegFormer is consistently better than SETR, Swin, PVT and CvT in mIoU. In fact, we have previously reported the table where SegFormer/CvT/Swin/PVT are all compared at 160K iterations. SegFormer shows considerable improvement over in terms of mIoU.
> > >
> > > | Encoder 	| Decoder    	| Training Iterations 	| mIoU 	| ImgNet Top-1  	| Enc. Params (M) 	|
> > > |---------	|------------	|:-------------------:	|:----:	|:-------------:	|:---------------:	|
> > > | MiT-B2  	| MLP (Ours) 	|         160K        	| 46.4 	|      81.4     	|       24.2      	|
> > > | CvT-13  	| MLP (Ours) 	|         160K        	| 43.7 	|      81.6     	|       20.0      	|
> > > | Swin-T  	| UperNet    	|         160K        	| 44.5 	|      81.2     	|       28.0      	|
> > > | PVT-S   	| SemFPN     	|         160K        	| 43.2 	|      79.8     	|       24.5      	|
> > >
> > >
> > > That being said, to make sure that the reviewer won’t have any further concerns, we will also report experiments with 160K training for all the other encoder/decoder combinations in the next few days.

---

> > > > ### Comment · Reviewer_n5uX · 2021-08-22
> > > > **Response**
> > > >
> > > > I thank the authors for their response and understand their concerns. I want to emphasize that I provide reviews not because I want to reject this paper, but to **help the authors to improve their work**. It looks like the authors are placing me as an "opponent" and not going to take any of my advice at all. But anyway, I will still answer the authors' questions.
> > > >
> > > > _**Re: "Self-contradicting comments" and "Unreasonably picky requirements"**_
> > > >
> > > > I'm not trying to say whether image classification accuracy is good or not, I'm asking what is specific design that makes the new model suitable for semantic segmentation. The authors pointed out a few points, I do not completely agree with all of them. The only thing that I think makes sense is removing positional embedding. However, for all these claims, if I change the subject from semantic segmentation to object detection, they still apply. That is why I keep asking why the authors position their paper on "semantic segmentation" if there is nothing specific to semantic segmentation.
> > > >
> > > > _**Re: "Selective/twisted interpretation and double standard"**_
> > > >
> > > > This is not related to the technical aspect. But if the authors really want to compare their paper with Mask R-CNN, I would like to ask does Mask R-CNN propose a new backbone? The Mask R-CNN is so simple to show the advantage of having a parallel mask head on top of Faster R-CNN. Does SegFormer have such a simple and clear message?
> > > >
> > > > _**Re: "Difference with CvT and PVT"**_
> > > >
> > > > Yes, CvT and PVT are concurrent work, but this does not mean we should not protect the ideas of concurrent works. As also pointed out by other reviewers (e.g., GD12), "The presented novelties ... are intermixed with techniques from prior work ... which does not merit a separate conference paper".
> > > >
> > > > I do not want to argue with the authors on these non-technical aspects anymore, I'm looking forward to the updated ablation results.

---

> > > > > ### Author Response · Authors · 2021-08-25
> > > > > **Rebuttal by Paper1675 Authors**
> > > > >
> > > > > We thank the reviewer again for the continued follow-up. This will also be our last open response containing non-technical issues.
> > > > >
> > > > > It is never our intention to treat anyone as “opponent” simply because of criticism. As authors, we understand that taking the reviews constructively is important and the openreview discussion platform was meant to be mainly discussing technical issues. However, it is also our right to raise valid concerns regarding fairness of the review, and we have been careful not to abuse it in the first place.
> > > > >
> > > > > It is untrue that we “don’t take any of the reviewer’s advice at all”. Our initial long responses have addressed many concerns from R3/R4. We particularly agreed with R3/R4 on some valid concerns such as clearer contributions and ablation on different combinations of encoder/decoder. We have taken them constructively with detailed answers and new experiments. Yet this does not mean that we should be forced to accept some of the controversial standards. Throughout the discussion we found that our disagreement mainly lie in **the “standard” to justify the contribution**:
> > > > >
> > > > > Our position in this paper is pretty well summarized by R1 in the follow-up review. The reviewer argues that our work is not segmentation-specific and listed many reasons, but the reasons are not convincing. For example, the reviewer insisted on the separate study of decoder and encoder while we have shown clear motivation and benefits of the coupled design in segmentation. To us such a standard remains questionable without reasonable justifications.
> > > > >
> > > > > The reviewer mentioned that the positional encoding free design is not segmentation-specific (applicable for detection). Again this standard seems a bit questionable because a clear counter example would be DeepLab where the concept of dilated convolution and contextual modeling clearly applies for detection as well (e.g., [a]). CvT is another good example since the mixed convolutional/self-attention design clearly makes sense to segmentation and detection problems as well but the paper focuses on image classification only. For us, a more self-contained standard would probably be **whether a design is shown to benefit certain task(s) significantly**. We agree with the reviewer that showing a generalized framework that benefits multiple tasks is certainly great. However, using this as a universal standard seems to defeat the purpose of pushing the field forward. Given the strong connection between many vision tasks nowadays (eg., segmentation vs depth estimation), it is not surprising that one design can find applications in other tasks. Such a standard would virtually require each work to provide a “proof by exhaustion”, and does not seem a constructive reason for rejection.
> > > > >
> > > > > More importantly, we feel that the reviewer is missing the following key big picture: If the ultimate concern is semantic segmentation, then nothing is more self-explanatory than the SOTA efficiency and performance which the semantic segmentation community greatly cares about. We have demonstrated the strong performance of SegFormer on semantic segmentation tasks. SegFormer’s backbone shows a reasonable ImageNet classification performance similar to Swin and CvT, but the semantic segmentation performance of these two backbones are relatively lower than SegFormer. This supports our claim that the proposed designs are indeed segmentation-friendly.
> > > > >
> > > > > [a] Li et al., Scale-aware trident networks for object detection, ICCV19
> > > > >
> > > > > **Mask R-CNN**
> > > > > We believe the message is clear. As answered in the previous response, our work simplifies the decoder design with higher efficiency while maintaining good performance. We are also the first to provide an explanatory framework for the decoder design through ERF analysis. We think this delivers a clear message/guidance to the future decoder design in semantic segmentation with Transformers.
> > > > >
> > > > > **Difference with CvT and PvT**
> > > > > We agree with the reviewer that the ideas of concurrent work should be protected. We like CvT and think it’s an elegant work deserving to be published at top venues. We think the best way is to acknowledge the credit from both works rather than using one to counter another. This is probably the true motivation of the conference policy.
> > > > >
> > > > >
> > > > >
> > > > >
> > > > > **160K results**
> > > > > Per request, we have conducted the 160K full training on the rest of encoder/decoder combinations (in addition to the table in our previous response). One could see that the conclusion is similar to the results from the 40K training.
> > > > >
> > > > >
> > > > >
> > > > > |    Encoder    	|     Decoder    	| Training Iterations 	| mIoU 	|  FPS 	| Dec. GFlops 	| Dec. Params (M) 	|
> > > > > |:-------------:	|:--------------:	|:-------------------:	|:----:	|:----:	|:-----------:	|:---------------:	|
> > > > > | MiT-B2 (Ours) 	| UperNet (Swin) 	|         160K        	| 46.5 	| 14.2 	|    210.7    	|       29.7      	|
> > > > > | MiT-B2 (Ours) 	|   MLA (SETR)   	|         160K        	| 46.2 	|  9.5 	|     87.7    	|       4.2       	|
> > > > > | MiT-B2 (Ours) 	|   MLP (Ours)   	|         160K        	| 46.5 	| 21.4 	|     42.1    	|       3.3       	|
> > > > > | MiT-B5 (Ours) 	| UperNet (Swin) 	|         160K        	| 50.7 	|  5.3 	|    210.7    	|       29.7      	|
> > > > > | MiT-B5 (Ours) 	|   MLA (SETR)   	|         160K        	| 50.9 	|  3.8 	|     87.7    	|       4.2       	|
> > > > > | MiT-B5 (Ours) 	|   MLP (Ours)   	|         160K        	| 51.0 	|  9.8 	|     42.1    	|       3.3       	|
> > > > > |     Swin-T    	|   MLP (Ours)   	|         160K        	| 43.4 	| 20.6 	|     42.8    	|       3.6       	|
> > > > > |     Swin-T    	| UperNet (Swin) 	|         160K        	| 44.5 	| 15.4 	|    211.3    	|       31.4      	|
> > > > > |  ViT-L (SETR) 	|   MLP (Ours)   	|         160K        	| 47.7 	|  4.7 	|     0.6     	|       0.6       	|
> > > > > |  ViT-L (SETR) 	|   MLA (SETR)   	|         160K        	| 47.7 	|  4.6 	|     1.8     	|       3.7       	|
> > > > > |     PVT-S     	|   MLP (Ours)   	|         160K        	| 43.2 	| 31.9 	|     5.5     	|       0.6       	|
> > > > > |     PVT-S     	|  SemFPN (PVT)  	|         160K        	| 43.0 	| 29.7 	|     21.4    	|       4.2       	|

---

> > > > > > ### Comment · Reviewer_n5uX · 2021-08-25
> > > > > > **Response**
> > > > > >
> > > > > > I thank the authors for their explanations.
> > > > > >
> > > > > > The 160K results ablation table is an important missing experiment and I strongly recommend the authors add these results in the **main text**.
> > > > > >
> > > > > > On the other hand, these numbers also suggest the main improvement comes from the **backbone** instead of the MLP decoder (the MLP decoder only contributes to the efficiency of SegFormer), and this is why I keep saying that I don't agree with the positioning of this paper: the authors could have focused on vision transformer backbone design (the MiT model).
> > > > > >
> > > > > > I will think about the authors' responses more carefully before I make the final recommendation.

---

> > > > > > > ### Author Response · Authors · 2021-08-26
> > > > > > > **Author Response**
> > > > > > >
> > > > > > > We will add the 160k ablation study into the main paper. The MLP decoder is also in line with our motives to build a simple, efficient but powerful segmentation framework.
> > > > > > >
> > > > > > > Though there have been some disagreements, we thank the reviewer and appreciate the constructive suggestions/patient discussion in this process. We hope the reviewer can kindly reconsider the previous decision.

---

> ### Author Response · Authors · 2021-09-09
> **Promise Draft Revision**
>
> Thank you for the kind final update. We are glad to see converged understandings and positive discussions in the end. Although there have been some misunderstandings and disagreements in the process, we sincerely appreciate the criticisms because they are essential for the improvement of the paper. We agree with the reviewer on many valid questions/concerns and will take them seriously.
>
> Our apology for the missing revised version. We were relatively late to notice the further update of comments (no email notification). There won't be major technical changes/difficulties to incorporate all the needed changes. We respect the criticisms and hereby promise the revision.
>
> A revised draft will also be attached soon to address the concerns. We will pay more attention to any concerns/follow-up comments regarding the revision.

---

> > ### Author Response · Authors · 2021-09-10
> > **Fully Revised Draft**
> >
> > Dear Reviewer n5uX, we have attached a revised draft following the suggestions. Hope this will address your final concerns.

---

### Official Review · Reviewer_GD12 · 2021-07-05

**Rating:** 6
**Confidence:** 4

**Summary:**

This paper introduces the SegFormer model, a Transformer-based model for semantic segmentation (i.e. dense pixel classification) in images. The model uses prior architectural innovations such as a pyramidal structure (progressive downsampling) and a factorized version of self-attention for computational efficiency. The main novelties over prior work on Transformer-based semantic segmentation models and image classification models are: 1) the use of convolutional filters to propagate position information from the image boundaries (as opposed to using explicit positional encodings) and 2) skip-connections between individual layers directly to the output layer, using spatial upsampling. Experimental evaluation demonstrates improved mIoU scores on default semantic image classification benchmarks compared to a baseline SETR (Segmentation Transformer) model, while achieving computational benefits thanks to the adoption of a more efficient Transformer architecture.

**Limitations And Societal Impact:**

Limitations are not clearly discussed, apart from the following limitation mentioned by the authors:
"One limitation is that although our smallest 3.7M parameters model is smaller than the known CNN’s model, it is unclear whether it can work well in a chip of edge device with only 100k memory. We leave it for future work.". -- It is not entirely clear how number of parameters relates to memory consumption, as the latter is often dominated by intermediate activations and gradients. It would be good to discuss limitations specific to this particular model class. What are interesting next steps for future work?

In terms of (negative) societal impact the authors write "We do not foresee obvious undesirable ethical/social impacts at this moment." -- I would encourage the authors to consider highlighting potential negative societal impact stemming from future applications by techniques such as theirs, e.g. in the area of surveillance.

**Main Review:**

The paper is overall well-written, well-structured and easy to follow, although clarity could be improved in several aspects (see below). The architectural choices are well-motivated and experimental results demonstrate clear advantages over prior approaches. The code / model implementation looks clean, and providing this implementation is a big plus and will allow for easy reproduction. Transformer architectures for applications in visual processing are very popular right now, and so the paper should be relevant for a NeurIPS audience, although the presented scope of the method (semantic pixel classification) is rather narrow.

Overall, despite the positive points mentioned above, I am arguing for rejecting this paper in its current form. I lay out the reasons for this decision in the following.

The presented novelties (1. use of convolutional filters to propagate position information from the image boundaries; 2. skip-connections between individual layers directly to the output layer) are intermixed with techniques from prior work, such as using a pyramidal Transformer architecture (as in [1]) and the use of an efficient, factorized form of self-attention (also from [1]). In the experimental section, it is unclear whether the performance benefits stem from utilizing these components from prior work [1] or whether the benefits stem from the novel contributions 1. and 2. -- the computational efficiency gains appear to be largely due to the use of the architecture from [1], which does not merit a separate conference paper (swapping out an image classification with a pixel classification target is of limited novelty).

The effective receptive field analysis and the analysis of robustness to natural corruptions are interesting, but they do not compare to the more recent SETR baseline, but instead to a weaker ConvNet-based baseline published in 2018. They thus unfortunately add little to convince the reader about the benefits of the proposed novelties over the current state of the art. The showcased qualitative improvements on Cityscapes in Figure 4 compared to SETR appear to be quite marginal (mask boundaries are marginally sharper, but otherwise unchanged). If this is a representative example of the obtained improvement, then it is unclear how this method will improve prediction quality in downstream applications.

The statement that the proposed "encoder can easily adapt to arbitrary test resolutions without impacting the performance" is not adequately verified. In fact, I would expect this to be false: if I increase the resolution 100x of the input image, I would very much expect the model to struggle (not just computationally, but also in terms of propagating position information throughout the model). The experiment in Table 1c suggests that increasing the resolution and aspect ratio by a small factor leads to only little loss in performance, but this does not justify the above statement.

Since the novel form of (implicit) positional encoding is a core contribution, it should be investigated more. For example, one could try to measure the model's ability to predict positional encodings by either: 1) giving the model PE as inputs (i.e. trivial auto-encoding) vs. 2) using the proposed position-free approach. This would allow for a good comparison of how 1) and 2) compare in terms of generalization to new image sizes.

Regarding the second novelty, i.e. skip-connections to the decoder: SETR introduces a related strategy termed "Multi-Level feature Aggregation (MLA)". A direct comparison against this technique (+ upsampling where necessary) would strengthen this contribution.

Clarity of the paper could be improved. There are many typos and grammatical mistakes, which should be easy to fix by doing another pass over the paper. The "Overlapped patch merging" strategy is described in a confusing way and parameters such as the padding size P are not explained. Looking at the code, this operation is simply implemented by a convolutional layer, and hence it would greatly simplify the description if the authors replaced this paragraph by simply saying that they use a convolutional layer after Transformer blocks, if I understand the method correctly.

Lastly, the impact of this method could be greatly increased by considering (and comparing on) other pixel-level prediction tasks such as depth prediction, which the architecture should support out-of-the box.


Other comments / questions, suggestions for improvement:
* The references [2,3] as cited in the sentence "In the deep learning era [...]" (Section 2, first sentence) look quite out of place compared to the other (seminal) works cited in this context. Were these cited by accident? I recommend explaining their relevance in the text (or dropping them).
* SETR uses SGD with momentum as optimizer whereas the present method uses AdamW. How much of the performance difference can be explained by the different choice of optimizers? Doing an ablation study over this choice would help clarify where the performance benefits come from (i.e. train with SGD and momentum in a setting similar to SETR).
* What is the unit for Params in Table 2?
* Why did the authors not compare on PASCAL Context (as in the SETR paper)?
* No error bars are reported and the authors state that the variance between seeds was low ("quite table mIoU results"). Would the authors be able to quantify this variance on at least one experiments, to get an idea whether results are significant?


[1] Wenhai Wang, Enze Xie, Xiang Li, Deng-Ping Fan, Kaitao Song, Ding Liang, Tong Lu, Ping Luo, and Ling Shao. Pyramid vision transformer: A versatile backbone for dense prediction without convolutions, 2021
[2] Xiang Li, Wenhai Wang, Xiaolin Hu, and Jian Yang. Selective kernel networks. In CVPR, 2019.
[3] Wenhai Wang, Xiang Li, Tong Lu, and Jian Yang. Mixed link networks. In IJCAI, 2018.


----- UPDATE AFTER REBUTTAL ----

I would like to thank the authors for their very extensive and detailed response to my review. My concerns about positioning of the paper still remain, i.e. it is difficult to clearly pinpoint what is the main contribution of this work, as the method clearly benefits from a careful combination of several architectural contributions in conjunction with careful optimization. In this point I largely agree with the concerns raised by reviewer n5uX.

At the same time, the authors have done an impeccable job at addressing my other concerns and at carefully investigating various aspects and components of the proposed model. Given the detailed experimental evaluation and the clear usefulness of this model as a strong baseline for the semantic segmentation community (as pointed out by reviewer LMud), I believe that this paper lies marginally above the acceptance threshold despite concerns around positioning and significance of the individual technical contributions.

**Time Spent Reviewing:**

4

---

> ### Author Response · Authors · 2021-08-10
> **Rebuttal by Paper1675 Authors**
>
> Dear Reviewer GD12,
> Thank you for the detailed review. We addressed some comments in the Author response to all reviewer and, below, we provide additional clarifications to your comments.
>
>
> **General Response:**
> We understand that two major concerns  are:
> (A) Unclear novelties and contributions
> (B) Lacking comparisons and analysis with SETR.
>
> For (A), we have listed the additional contributions beyond PVTv1 in **Difference between SegFormer and PVTv1** in the **Author response to all reviewers**. We note that Reviewer n5uX has also raised concerns/questions to these additional contributions. We believe there are misunderstandings that caused these concerns, and we address them below point by point.
>
> For (B), we have added new experiments covering SETR on Cityscapes-C and SegFormer on PASCAL-Context. We also address other concerns below point by point.
>
> **Q1 Novelties and whether the performance/efficiency benefits from PVTv1.**
> A1: We have clarified this in our general response to all reviewers. Please kindly check our answer in **Difference between SegFormer and PVTv1**. We feel that a potential misunderstanding here is that designs from PVTv1 are indeed necessary to achieve high efficiency, but are not enough to guarantee the SOTA efficiency-performance pareto-frontier of SegFormer.
>
> This is clearly verified in our ablation studies with different combinations of encoders/decoders, as well as Fig. 1 in the main paper. The large improvement indicates that several novel designs not presented in PVTv1, including:
> 1) The positional embedding free design
> 2) The simple MLP decoder design
> 3) Overlapped patch embedding
>
> can significantly benefit segmentation problems. The conclusion “... swapping out an image classification with a pixel classification target is of limited novelty” therefore fails to correctly capture this work.
>
> **Q2: Missing effective receptive field analysis (ERF) with SETR.**
> A2: First, we separate our response of ERF from that of the corruption experiment, because these two items serve for different purposes and are therefore motivated differently.
>
> A fundamental misunderstanding here probably lies in misinterpreting the motivation of ERF analysis: The simple MLP design is an important design that promotes both efficiency and performance, and we thus want to know why such a simple design is successful. Even more interestingly, this design seems to be Transformer-specific, but is less effective on CNNs. The ERF analysis serves as an explanatory framework that provides insights to this phenomenon. Therefore, choosing a representative CNN method is most important for us, not choosing an absolute SOTA (Transformer) method.
>
> It is true that our ERF analysis on other Transformer based methods, such as SETR, may lead to similar observations. But this may be a bit orthogonal to our motivation, and does not change ERF being a contribution of this work.
>
> **Q3: Missing comparison with SETR on Cityscapes-C.**
> A3: Thanks for this suggestion. Following the suggestion, we benchmarked SETR-DeiT (https://github.com/fudan-zvg/SETR/blob/main/configs/SETR/SETR_MLA_DeiT_768x768_40k_cityscapes_bs_8.py) model that is also pretrained on ImageNet-1k. The results are as follows:
>
> | method       	| Gaus Noise 	| Shot Noise 	| Impl Noise 	| Spec Noise 	| Moti Blur 	| Defo Blur 	| Glass Blur 	| Guas Blur 	| Snow 	| Spat 	|  Fog 	| Forst 	| Bright 	| Contr 	| Satu 	| JPEG 	|
> |--------------	|:----------:	|:----------:	|:----------:	|:----------:	|:---------:	|:---------:	|:----------:	|:---------:	|:----:	|:----:	|:----:	|:-----:	|:------:	|:-----:	|:----:	|:----:	|
> | SegFormer-B5 	|    57.8    	|    63.4    	|    52.3    	|    72.8    	|    69.1   	|    68.6   	|    64.1    	|    69.8   	| 40.7 	| 68.4 	| 78.5 	|  49.9 	|  81.0  	|  77.7 	| 80.1 	| 58.8 	|
> | SETR-DeiT    	|    54.7    	|    60.5    	|    51.9    	|    69.4    	|    64.9   	|    65.1   	|    59.1    	|    65.3   	| 44.3 	| 64.8 	| 68.2 	|  39.1 	|  74.9  	|  69.6 	| 74.9 	| 58.5 	|
>
>
>
> It should be mentioned that SETR is a very nice method with strong robustness as well, probably also benefiting from the self-attention design. SegFormer outperforms the SETR baseline in general except for one corruption (snow). We will also include the above results in the revised version of the paper.
>
> That being said, we want to emphasize that SegFormer is the first work to benchmark a transformer architecture on segmentation robustness. By the time we submit this work, the best and latest reference we could find on this problem is [a] where our results, baselines, and the entire settings are cited from. We believe that the observation of strong zero-shot robustness of Transformers over CNNs should be part of our contributions.
>
> [a] Kamann et al., Benchmarking the Robustness of Semantic Segmentation Models, CVPR 2020.
>
> **Q4: The showcase compare with SETR is marginal, the improvement only on boundary.**
> A4: We believe that the conclusion is not true here. In fact, boundary qualities do matter as several works [b, c] have pointed out. In particular, [b] has shown that most of the IoU error is distributed within a few pixels of the boundary. This may be a bit counter intuitive to the general perception that long-range errors contribute more to the overall errors, as the reviewer is probably expecting. On the other hand, we have included additional visualization results in the Appendix and shown that SegFormer outperforms SETR not only on the object boundary but also inside the object.
>
> [b] SegFix: Model-Agnostic Boundary Refinement for Segmentation, ECCV 2020.
> [c] PointRend: Image Segmentation as Rendering, CVPR 2020.
>
> **Q5: Arbitrary image size under extreme cases, e.g., with 100x increased resolution.**
> A5: We apologize if the term “arbitrary test resolutions” is causing any confusion. We certainly agree with the reviewer that any method would fail under extreme cases. We feel that this is probably a communication issue. From the context of our paper, we intend to target *certain level of* training/testing crop size mismatch, mostly because of many common segmentation issues, such as the limitation of current GPU memories (which prevents one from training with ultra-large crops) or the requirement of multi-scale testing. In these cases, the scaling factors often lie in the range of [0.5, 2.0]. An extreme case with 100x increased resolution as a counter example may be meaningless (even for super-resolution problems).
>
> We do observe significant improvement from our design under the mild training/testing resolution mismatch mentioned above. On the other hand, SETR is restricted to sliding window based inference, because of its fixed-shape positional embedding. PVT tries to interpolate PE to the image size but the performance is also less competitive.
>
>
> **Q6: More Investigation about Positional Encoding.**
> A6: We are not entirely sure whether we have understood R3’s motivation correctly here. To verify what R3 suggested, we conduct an ablation study on ADE20K where we use CPVT to generate the conditional positional encodings (CPE) instead of using Mix-FFN. We find that the performance of Mix-FFN is slightly better, which shows that the adding operation in CPE may not be necessary.
>
> | method  	| mIoU 	|
> |---------	|:----:	|
> | CPE    	| 44.3 	|
> | Mix-FFN 	| 45.4 	|
>
> **Q7: Comparison with SETR’s MLA decoder.**
> A7: Thanks for the suggestion. We replaced the decoder in SETR with MLP. We find that SETR+MLP achieves slightly better performance than SETR, faster speed and lower FLOPs due to the simpler decoder design. The experiment does show the importance of decoder design to achieve better efficiency-performance results.
>
> | Encoder       	| Decoder    	| mIoU 	|  FPS 	| Decoder GFLOPs 	| Decoder Params (M) 	|
> |---------------	|------------	|:----:	|:----:	|:-----------:	|:---------------:	|
> | MiT-B2 (Ours) 	| MLA (SETR) 	| 45.2 	|  9.5 	|     87.7    	|       4.2       	|
> | MiT-B2 (Ours) 	| MLP (Ours) 	| 45.4 	| 21.4 	|     42.1    	|       3.3       	|
>
>
> **Q8: Improved explanation of overlapped patch embedding and padding size P.**
> A8: We apologize for the confusion. Both the implementations of patch embedding in ViT and our paper are conv2d, and we mostly followed ViT to implement and name the operator. The padding size is also a parameter of Conv2D. We will improve the explanation in the final version following R3’s suggestion.
>
> **Q9: Inappropriate Reference.**
> A9: Thanks. We will remove [2,3] in future versions as suggested.
>
> **Q10: Performance on different optimizers.**
> A10: We believe that different methods tend to have different sets of optimal training recipes, including the most suitable optimizers. SegFormer uses AdamW as many methods did [d, e, f]. We assume that the authors of SETR find SGD the best optimizer for SETR. To verify this, we compare SETR training with both SGD and AdamW. We tried hard to tune for the best performance of SETR + AdamW. However, we still find that that SGD works better than AdamW on SETR.
>
> | method 	| Optimizer 	| Training Iterations 	| mIoU 	|
> |--------	|-----------	|---------------------	|------	|
> | SETR   	| SGD       	| 40K                 	| 44.4 	|
> | SETR   	| AdamW     	| 40K                 	| 41.3 	|
>
>
> For SegFormer, using AdamW is better than using SGD.
>
> [d] Swin Transformer: Hierarchical Vision Transformer using Shifted Windows, arxiv21
> [e] An Image is Worth 16x16 Words: Transformers for Image Recognition at Scale, ICLR 2021
> [f] Pyramid Vision Transformer: A Versatile Backbone for Dense Prediction without Convolutions, arxiv21

---

> > ### Author Response · Authors · 2021-08-10
> > **Rebuttal by Paper1675 Authors**
> >
> > **Q11: Unit for Params in Table 2.**
> > A11: The unit for parameters is million (M). We will clarify this in the revised version of the paper.
> >
> >
> > **Q12: Why not compare with SETR on the Pascal Context dataset.**
> > A12: We have reported results on three major datasets. Considering the limited space of the paper, we did not train the model on the Pascal Context dataset. As requested by the reviewer, we have trained SegFormer on Pascal Context and, as shown below, at a smaller model size, SegFormer yields competitive results compared to SETR. We did not have enough time to thoroughly optimize the model on this task. We will add this result to the revised version of the paper.
> >
> > | method       	| mIoU (MS) 	| FPS 	| Params (M) 	|
> > |--------------	|:---------:	|:---:	|:----------:	|
> > | SETR         	|    55.8   	| 5.5 	|    309.3   	|
> > | SegFormer-B5 	|    56.1   	| 6.5 	|    84.6    	|
> >
> > **Q13: Report error bars to state the result is stable.**
> > A13: We repeated our experiments 5 times on ADE20K, training with 40k iterations and evaluating for every 1k iteration. We observe that the variance of mIoU is smaller than 0.5% in terms of  the highest result. We will add these numbers in the revised version of the paper.
> >
> > That being said, we mostly just followed the traditional settings in many previous works [a, b, c], where they also do not report error bars. More importantly, we observed that the results are relatively stable.
> >
> > **Q14: Limitations and social impact**
> > A14: We thank the reviewer for pointing this out, and apologize for some of the unrigorous descriptions. In our case, we mostly consider inference so gradient is not part of our concerns. However, the reviewer is correct in pointing out that activations will also consume memories. Since an edge device needs to store both the parameters and activations, the 3.7M of our lightest model is a lower bound of the memory requirement. We will make it clearer as the reviewer suggested.
> >
> > For the specific next steps, we consider that even our lightest model may still be too heavy for some edge devices. Thus mixed-precision training, pruning, hardware-friendly self-attention designs and energy consumptions are important parts of our future work.
> > As work targeting autonomous driving/robotics applications, safety is always a main consideration. Although SegFormer leads to significantly improved zero-shot robustness, we fully understand this work is by no means perfect and there will still be many challenges towards reliable real world application. Our models may be subject to biases and other possible undesired mistakes, depending on how they are trained in reality. We didn’t focus on potential negative impacts on surveillance because this work was not mainly designed for surveillance applications. But yes, as a recognition model it may be used for surveillance applications, just similar to many other general methods such as object detection.
> >
> >
> > [a] Rethinking Semantic Segmentation from a Sequence-to-Sequence Perspective with Transformers, CVPR2021
> > [b] Encoder-Decoder with Atrous Separable Convolution for Semantic Image Segmentation ECCV2018
> > [c] Pyramid Scene Parsing Network, CVPR2017

---

> > > ### Comment · Reviewer_GD12 · 2021-08-26
> > > **Response to authors**
> > >
> > > Thank you for your very detailed response to my review. Please see my updated review for my response.

---

> > > > ### Author Response · Authors · 2021-08-29
> > > > **Author Response**
> > > >
> > > > We thank the reviewer again for the detailed discussions and the kind support of this work. Your constructive feedback and criticisms will help us greatly towards improving this work.

---

### Official Review · Reviewer_vMMz · 2021-07-10

**Rating:** 6
**Confidence:** 5

**Summary:**

This paper presents a simple, efficient yet powerful semantic segmentation framework which unifies Transformers with lightweight multilayer perceptron (MLP) decoders. The proposed SegFormer has two appealing features: 1) SegFormer comprises a novel hierarchically structured Transformer encoder which outputs multiscale features. It does not need positional encoding, thereby avoiding the interpolation of positional codes which leads to decreased performance when the testing resolution differs from training. 2) SegFormer avoids complex decoders. The extensive experiments show that the proposed method achieves the state-of-the-art performance on semantic segmentation task.

**Ethical Concerns:**

No Ethical Concerns

**Limitations And Societal Impact:**

the suggestions are presented in Main Review part

**Main Review:**

1. The paper presents a transformer based network for semantic segmentation task. Different from the last transformer based network SETR, the proposed SegFormer uses  positional-encoding-free,  hierarchical Transformer encoder and lightweight All-MLP decoder, which is more efficient and flexible than SETR.
2. My concerns and suggestions about this paper are as follows:
         1). The author claims that the segformer is novel, but the used overlapped patch embedding, efficient attention, the multi-stage structure are basically from PVT, PVTv2.  There is no any novel module or techniques proposed in this paper. So i think the novelty is limited.
         2). The author mainly uses the SETR as baseline method, but actually recent Swin and PVT (cited in the paper) also show the SOTA performance, which are not compared in the experimental report.
         3). The lightweight sseg models are compared in the experimental part, but only mobilenet based networks are listed,   ESPNet, ESPNetv2, BiSeNet, ICNet and other lightweight models are not reported and compared. i recommend the author can give more detailed comparison.
        4). In addition, UNet-like network is also popular network in semantic segmentation. Does it work for the transformer based network? it's better to give experiments about this.





**Time Spent Reviewing:**

5

---

> ### Author Response · Authors · 2021-08-10
> **Rebuttal by Paper1675 Authors**
>
> Dear Reviewer vMMz,
> Thank you for appreciating our approach. We address your comments below.
>
> **Q1: Designs borrowed from PVTv1/PVTv2.**
> A1: There may be a slight misunderstanding here. Please note that PVTv2 is an arXiv paper, which was released after our SegFormer submission. Therefore, it is not true that “the structure is from PVTv2”. Please refer to the more detailed answer to **Difference between SegFormer and PVTv1** in the **Author response to all reviewers**.
>
> **Q2: Compare with more real-time models e.g. ESPNet, ESPNetv2, BiSeNet, ICNet on Cityscapes.**
> A2: Thanks for the suggestion. In Table 2 of the main paper, we did compare with ICNet which we consider a highly representative method in real-time semantic segmentation. We also add comparisons to ESPNetv2 and BiSeNet in the table below. Note that BiseNet-X39 yields better FPS as it is optimized using TensorRT. We will add and discuss these results in the revised version of the paper.
>
> | method       	| mIoU 	| Params (M) 	| FPS              	|
> |--------------	|:----:	|:---------:	|------------------	|
> | BiseNet-X39  	| 69.0 	|    5.8    	| 105 (with TensorRT) 	|
> | ESPNetv2     	| 66.2 	|    3.5    	| -                	|
> | SegFormer-B0 	| 71.7 	|    3.8    	| 47.6             	|
>
> **Q3: Can U-Net like networks work with Transformers?**
> A3: We are not entirely sure whether we have captured the motivation here. We believe that U-Net, or other similar U-shape architectures can work with Transformers. U-shape architecture is also an encoder-decoder architecture, where the feature maps are progressively downsampled in the encoder, and progressively upsampled in the decoder. Our SegFormer with MLP decoder can be broadly regarded as a simple U-shape architecture. In addition, SegFormer with the decoder of Semantic FPN is also a typical U-shape architecture. We thus add an experiment on ADE20K using the Semantic FPN decoder. The result is similar and is shown in the following table:
>
>
> | Encoder       	| Decoder      	| mIoU 	|
> |---------------	|--------------	|:----:	|
> | MiT-B2 (Ours) 	| MLP (Ours)   	| 45.4 	|
> | MiT-B2 (Ours) 	| Semantic FPN 	| 45.0 	|

---

### Official Review · Reviewer_LMud · 2021-07-14

**Rating:** 7
**Confidence:** 5

**Summary:**

This work proposed an efficient Transformer-based semantic segmentation model called SegFormer. In essence, it consists of a hierarchical Transformer encoder and a lightweight MLP decoder. Compared to ViT, the transformer encoder employs overlapped patch merging, self-attention with reduced (key, value) for efficient computation, and a depthwise convolution inserted between two MLP layers in FFN to replace position embedding. The authors also conducted an analysis based on effective receptive field and revealed that transformer's encoder with large receptive field enables relative lightweight decoder design. Extensive experiments demonstrate that SegFormer achieves strong speed-accuracy trade-off and better robustness.

**Limitations And Societal Impact:**

Please refer to "Main Review" section above.

**Main Review:**

- Overall writing is clear and easy to follow. As for the method, the architectural design is simple yet effective as proven by extensive experiments and analysis. As a result, this could serve as a new baseline for semantic segmentation task.
- In L165, it was stated that "CPVT [54] uses 3x3 Conv together with PE to implement a data-driven PE". To my best understanding, [54] does not require PE. Instead, similar to this work, a 2D Conv with zero padding is employed to replace PE.
- Swin Transformers [9] is missing from the comparison with state-of-the-arts.
- The results in ablation experiment (Tab.1(b)(d)) does not match the final performance in Tab.1(a). For example, in Tab.1(d), the model variant MiT-B3 (S1-4) achieves 48.6 mIoU on ADE20K whereas MiT-B3 in Tab.1(a) attains 49.4/50.0 mIoU on the same ADE20K dataset despite having the same FLOPs and number of parameters. Why is this so?
___________
UPDATE AFTER REBUTTAL

I appreciate the authors' detailed rebuttal. Here are my responses after reading other reviewers' comments and the authors' response:

I have to admit that when I first read this paper, I had the same concern as other reviewers that this work shares high similarity with some recent vision transformer works, especially PvT [8] in terms of its hierarchical representation structure and efficient self-attention design. However, the efficient self-attention design was not claimed as the contribution of this work. In fact, the authors admitted in L155 that such design was taken from [8]. Nevertheless, I agree with reviewer GD12 that some of the presented components (e.g. overlapped patch merging, hierarchical representation) are intermixed with techniques from prior works.

The main reason for me to recommend an accept are two-fold: 1) the effectiveness of the proposed Mix-FFN for addressing train-test resolution mismatch; 2) the simplicity and efficiency of the overall design. The former is similar to LocalViT [57] which employs a 3x3 depthwise conv between two consecutive MLPs in FFN. Nevertheless, [57] retains the use of position encoding. On the other hand, this work demonstrates their Mix-FFN not only removes the need of positional encoding, but also fixes train-test resolution disparity while a fixed shape positional encoding fails to. I believe this finding is important for dense prediction task such as semantic segmentation where the testing resolution might differ from the training ones.

Secondly, the simplicity and efficiency of the design makes it serve as a strong baseline for semantic segmentation. SETR has previously shown that transformer backbones work well on semantic segmentation. Differently, this work demonstrates a compact and efficient solution for the task while performing comparably or even better. Although each component may not be entirely novel or inspiring, combining them to offer a simple yet effective solution is beneficial to the community, similar to the role of DeepLab series is playing in the segmentation community.

After reading other reviewers' comments and authors' response, I think most of the concerns were well addressed or at least to a certain extent (e.g. different combinations of encoder and decoder, comparison with more real-time segmentation models, variance of different runs, experiments on PASCAL Context dataset etc.). I agree with reviewer n5uX that the technical contribution may be insufficient for both vision and semantic segmentation community. However, as explained earlier, although each component may not be entirely novel, I think offering a feasible compact solution that could serve as a strong baseline for semantic segmentation would be beneficial to the community. After taking all these into accounts, I have decided to lower my rating to 7.

**Time Spent Reviewing:**

3

---

> ### Author Response · Authors · 2021-08-10
> **Rebuttal by Paper1675 Authors**
>
> Dear Reviewer LMud,
> Thank you for appreciating our approach. We will address your comments below.
>
>
> **Q1: Does CPVT [54] require positional encoding?**
> A1: CPVT does require positional embeddings (PE), but it generates conditional positional encodings (CPE) in a different way compared to the original ViT implementation. Both CPVT and ViT involve positional information by adding PE to the feature map, but they differ in how to generate PE.
> For CPVT, given an input feature X, it generates CPE with a 3x3 conv shown as below:
>
> $CPE=\mathtt{conv}_{3\times3}(X)$
> $X=X+CPE$
>
> Our work, on the other hand, *completely removes the PE.* The 3x3 conv in our Mix-FFN can already encode positional information and, thus, the addition operation in CPVT is no longer needed. Conceptually, our work goes one step further than CPVT towards removing PE. We have also verified that the proposed design does improve over the CPE design from CPVT. Please kindly refer to our response to R3 on **More Investigation about Positional Encoding**.
>
> **Q2: Results in Table 1(b) and (d) do not match those in Tab.1(a).**
> A2: The results mismatch comes from the different number of training iterations. In Table 1(a), we reported the results with 160k iterations. Table1(b) and (d) report ablation studies where we set the training iterations to 40K for all methods. There are no significant differences in the conclusions between the different number of iterations. We will make it clear in the revised version of the paper.

---

> > ### Comment · Reviewer_LMud · 2021-08-23
> > **Response to Authors Rebuttal**
> >
> > Dear authors,
> >
> > Thank you for the thorough response. I realized that updated review would not be notified via emails. Please refer to the review above for my updated comments.

---

> > > ### Author Response · Authors · 2021-08-25
> > > **Author Response**
> > >
> > > Dear Reviewer 1
> > >
> > > We sincerely thank the reviewer for the constructive feedback and support.

---

### Author Response · Authors · 2021-08-10
**Author Response to All Reviewers**

Dear all reviewers:

We sincerely appreciate the reviewers for the time and efforts on the review. We first address some common questions, followed by detailed responses to each reviewer separately. We hope our responses clarify existing doubts. We will really appreciate it if R3 and R4 can kindly reconsider the decision, provided that the main comments are well addressed.


**Missing comparison to Swin Transformers and PVT (R1/R2/R4)**
In fact, we compared SegFormer with multiple recent Transformer works, including PVT, Swin Transformer, and Twins and shown in Fig. 1 in the main paper. We thank the reviewers for reminding and will also update the results in Table 1 in the main paper.


**Difference between SegFormer and PVTv1 (R2/R3/R4)**
We notice that there is a general confusion between SegFormer and PVTv1 in terms of
the technical components and contributions. We understand this confusion and agree with the connections to PVTv1 regarding the hierarchical structure and the efficient self-attention (subsampling K/V). However, the hierarchical structure and efficient-self-attention designs in PVTv1 merely bring SegFormer to the same level of efficiency and performance of PVTv1, but not to the current state of the art performance. As shown in Fig. 1 of the main paper, there is a considerable gap between PVTv1 and SegFormer. SegFormer brings novelty from three different aspects:.

1) The positional-encoding-free design which reduces the sensitivity to resolution mismatch in training and inference. This feature is important for segmentation where resolution mismatch in training and inference is common.
2) A clean and simple decoder design that reduces the computation complexity while achieving good performance. We also provide a detailed analysis on why our decoder is suitable for Tansformers but it is not so effective for CNNs.
3) Overlapping patch merging is also a novel design over PVTv1

We believe these contributions are significant and necessary to achieve the state of the art performance of SegFormer, as clearly verified by the subsequent ablation studies.

We will make these differences clearer in the revised version of the paper.

**New experiments on ADE20K with different encoder/decoder combinations (R3/R4)**
Per request by R3 and R4, we conducted additional studies on different combinations of encoder/decoder architectures, with the results listed in the following table. We also address specific questions/concerns regarding this study. Please refer to our separate response to each reviewer.

| Encoder       	| Decoder        	| Training Iterations 	| mIoU 	|  FPS 	| Decoder GFLOPs 	| Decoder Params (M) 	|
|---------------	|----------------	|:-------------------:	|:----:	|:----:	|:-----------:	|:---------------:	|
| MiT-B2 (Ours) 	| UperNet (Swin) 	|         40K         	| 45.3 	| 14.2 	|    210.7    	|       29.7      	|
| MiT-B2 (Ours) 	| MLA (SETR)     	|         40K         	| 45.2 	|  9.5 	|     87.7    	|       4.2       	|
| MiT-B2 (Ours) 	| MLP (Ours)     	|         40K         	| 45.4 	| 21.4 	|     42.1    	|       3.3       	|
| Swin-T        	| MLP (Ours)     	|         40K         	| 41.4 	| 20.6 	|     42.8    	|       3.6       	|
| Swin-T        	| UperNet (Swin) 	|         40K         	| 42.5 	| 15.4 	|    211.3    	|       31.4      	|
| ViT-L (SETR)  	| MLP (Ours)     	|         40K         	| 44.7 	|  4.7 	|     0.6     	|       0.6       	|
| ViT-L (SETR)  	| MLA (SETR)     	|         40K         	| 44.4 	|  4.6 	|     1.8     	|       3.7       	|
| PVT-S         	| MLP (Ours)     	|         40K         	| 42.2 	| 31.9 	|     5.5     	|       0.6       	|
| PVT-S         	| SemFPN (PVT)   	|         40K         	| 41.9 	| 29.7 	|     21.4    	|       4.2       	|


**Other new experiments on ADE20K**
 * SegFormer with Semantic FPN decoder (U-shape architecture)
 * Performance of SETR on Cityscapes-C
 * Comparison between CPVT vs. Mix-FFN
 * SETR with different optimizers
 * Performance of SegFormer on the Pascal Context Dataset
 * Additional segmentation baseline with CvT backbone on ADE20K

**Other concerns**
Other concerns on typos, clarity, and figures will also be carefully addressed.

---

### Author Response · Authors · 2021-09-10
**Fully Revised Draft**

Dear Reviewers and ACs:

Thank you very much for the helpful reviews. We are thankful for the thorough suggestions on our previous manuscript. We have taken all the suggestions and made major changes to our previous draft, with the main changes marked red in the draft. Our final versions will be based on the following attached draft:

https://drive.google.com/file/d/1hMW645TXdHUnJSOsPPvcf5JmniIbx4iZ/view?usp=sharing

We have specifically made the following changes:

**In the main paper:**

1) Made the relation between SegFormer and PVT clearer and highlighted the contributions from us.
2) Added discussion on the relation and difference of our PE-free design with CvT.
3) Added the results of SETR on Cityscapes-C.
4) Revised the expression of "Overlap patch merging".
5) Removed unnecessary references.
6) Added new experiments on ADE20K with different encoder/decoder combinations.
7) Revised the limitations and broader impact of SegFormer.
8) Fixed typos.

**In the supplementary material:**

1) Added new experiments of Pascal Context dataset.
2) Compared with SegFormer with Swin/CvT/PVT.

Thank you very much again!

Best regards,
Paper 1675 Authors

---

### Decision · Program_Chairs · 2021-09-27

**Decision:**

Accept (Poster)

**Comment:**

The authors propose an efficient semantic segmentation model composed of a hierarchical Transformer encoder and a lightweight MLP decoder. The transformer encoder employs overlapped patch merging, efficient self-attention module, and a depthwise convolution inserted between two MLP layers in FFN to replace positional embeddings. The empirical evaluation demonstrates better accuracy/speed tradeoffs compared to competing models.
The paper was reviewed by 4 experts who appreciated the clarity of exposition, simplicity of the design, the effectiveness of the proposed approach in dealing with test-train resolution mismatch, as well as competitive performance. During the discussion the reviewers pointed out several concerns related to positioning and the significance of technical contributions which were mostly addressed. Finally, the reviewers agreed that the approach is a simple and competitive baseline for the semantic segmentation community. I will recommend acceptance.